# What's Producible May Not Be Reachable: Measuring the Steerability of Generative Models

**Keyon Vafa**
Harvard University

**Sarah Bentley**
MIT

**Jon Kleinberg**
Cornell University

**Sendhil Mullainathan**
MIT

## Abstract

How should we evaluate the quality of generative models? Many existing metrics focus on a model's *producibility*, i.e. the quality and breadth of outputs it can generate. However, the actual value from using a generative model stems not just from what it can produce but whether a user with a specific goal can produce an output that satisfies that goal. We refer to this property as *steerability*. In this paper, we first introduce a mathematical decomposition for quantifying steerability independently from producibility. Steerability is more challenging to evaluate than producibility because it requires knowing a user's goals. We address this issue by creating a benchmark task that relies on one key idea: sample an output from a generative model and ask users to reproduce it. We implement this benchmark in user studies of text-to-image and large language models. Despite the ability of these models to produce high-quality outputs, they all perform poorly on steerability. These results suggest that we need to focus on improving the steerability of generative models. We show such improvements are indeed possible: simple image-based steering mechanisms achieve more than 2x improvement on this benchmark.

## 1 Introduction

There is a wedge between how we evaluate generative models and how we intend to use them. For example, a common way to evaluate image generation models is to measure the quality of outputs they can generate, e.g. by measuring how realistic or diverse the images they produce are [70, 32, 46]. But when these models are used by people, their success depends on more than just what they're capable of producing: can users create the images they actually want?

In this paper, we introduce methods for quantifying model *steerability*: how well can a user with a specific end goal guide a model to achieve that goal? Evaluating and benchmarking steerability is necessary for improving the real-world performance of generative models, in the same way that producibility benchmarks have been crucial for improving the quality of their outputs [19]. But creating a benchmark for steerability is challenging for two reasons. First, existing methods for evaluating models under human use [48, 39, 81] blend steerability and producibility: if a model doesn't produce the image a user wants, is it because it's incapable of producing it or because it can't be steered? Second, measuring steerability requires knowing a human user's goals, which are often latent or difficult to articulate.

We first provide a mathematical framework for modeling steerability and producibility. The key insight of this framework is to decompose model performance into two terms: a producibility term (how well can models produce the kinds of outputs a human may want), and a steerability term (how well can models be steered by humans towards the best output they're capable of producing).

We then propose a procedure that resolves the core challenges of benchmarking steerability. The procedure: sample an output from a model and instruct users to steer the model toward that output. This procedure is straightforward and can be applied to a variety of generative models. For example, for

39th Conference on Neural Information Processing Systems (NeurIPS 2025).

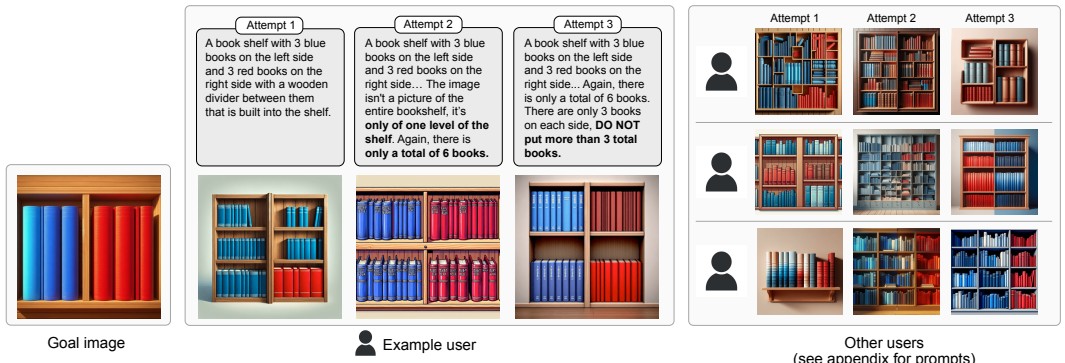

**Figure 1:** A goal image and attempts by users to steer toward it. To ensure the goal image is producible, it is sampled from the same model used for steering (here, DALL-E 3). Bolding added to prompts for emphasis. See Figure 10 for the full prompts.

text-to-image models, it calls for showing humans an image sampled from a model and then instructing them to prompt that same model to reproduce the image. This procedure addresses the two key challenges: first, as suggested by the framework, it evaluates a generative model by how well humans can steer it towards *an output it is capable of producing*. Second, to overcome the lack of access to a user's goal, it induces a goal by instructing users to get as close to as possible to a provided output.

We implement this benchmark in a large-scale user study for two domains: text-to-image models and large language models. Across the board, we find that the steerability of models is poor, both for normal participants and professional prompt engineers. For the image models, human annotators rate the attempted reproductions as dissatisfactory 60% of the time. Moreover, attempts to refine prompts do not reliably improve outputs; after five tries, the final image is closer to the goal image only 62% of the time (compared to a base rate of 50%).

Like many metrics, it is easy to optimize for steerability alone. However, this could come at the expense of producibility. To assess whether differences between models reflect true steerability improvements, we provide another mathematical decomposition that disentangles gains from better steering mechanisms and gains from producibility differences. This decomposition requires a predictive model of steerability; we show that steerability can be predicted with machine learning methods, and use it to analyze models, finding that steerability differences reflect both true improvements in steering mechanisms along with differences in producibility.

Finally, we show that the problem of poor steerability is addressable. We consider a simple steering mechanism that uses two ingredients: First, after an initial text prompt, we enable users to steer via suggested images rather than by prompt rewrites. Second, rather than suggesting images randomly, we use an auxiliary model to suggest images. Despite the simplicity of this technique, it achieves more than 2x improvement over text steering.

## 2 Framework

We define a *generative model m* over a domain $X$ to have two components: a *producible set* $S_m \subseteq X$ and a *steering mechanism* that allows humans to produce an instance in $S_m$. For example, the producible set of a text-to-image model is all the images the model can generate, and its steering mechanism is the text interface that humans use to guide generations.[1]

How do humans interact with the generative model? We assume that a human is trying to generate an instance to satisfy a specific use case. For example, someone using an image generation model may be looking for a certain kind of image, and would try to prompt the model to produce such an image. To model a specific user's use case, we define a family of reward functions $\mathcal{R}$ where each $r \in \mathcal{R}$ is a function over instances $r : X \to \mathbb{R}$. Each use case is defined by a single reward function.

---

[1]We can also consider distributions over producible instances rather than a single producible set without loss of generality; we use set notation here for simplicity but will also consider distributions in Section 4.

A human interacts with a generative model with the goal of producing an output that maximizes their reward. We define $h(m, r) \in S_m$ as the instance that is produced when a human with reward function $r$ interacts with the model $m$. We refer to $h$ as the *steering function*; it describes how someone with a specific reward function would steer a model to maximize it.[2] This notation abstracts away details about how interactions are constructed (e.g. the number of attempts users have to prompt a model). While we exclude these details in our notation, our experiments will explicitly consider different settings.

A human's reward when using a generative model is $r(h(m, r))$. This quantity describes how effective a generative model is under human interaction. However, it blends a model's steerability with the quality of its producible set $S_m$. To see this, we introduce additional notation to decompose the reward. For a set of instances $A$, define $r_A^*$ as the maximum reward when constrained to instances in $A$:

$$r_A^* = \max_{x \in A} r(x). \tag{1}$$

Denoting by $r_X^*$ the maximum reward over all possible instances, the largest possible reward for a given reward function is $r_X^*$. A model's efficacy can be summarized by the gap between this largest possible reward and the model's reward: $r_X^* - r(h(m, r))$. This gap can be decomposed:

$$r_X^* - r(h(m, r)) = \overbrace{r_X^* - r_{S_m}^*}^{\text{producibility gap}} + \overbrace{r_{S_m}^* - r(h(m, r))}^{\text{steerability gap}}. \tag{2}$$

The first term, $r_X^* - r_{S_m}^*$, is the *producibility gap*; it captures how well the producible set of a model aligns with the set of all possible instances $X$, regardless of how steerable the model is. Meanwhile, the second term is the *steerability gap*; it describes how well a model can be steered by humans *towards the best instance the model is capable of producing*.

The decomposition in Equation 2 describes a single reward function; in practice, a model's total reward will be averaged over all reward functions that humans may have. Thus, the producibility gap can equivalently be characterized by considering only the set of instances $T$ that humans may want to produce, i.e. those that maximize a feasible reward function (so that $r_X^* = r_T^*$ for all $r$). This decomposition reveals a couple of results. The first is that with perfect steering, a model's performance can simply be evaluated by comparing the producible set $S_m$ to the set of instances people may want to produce, $T$. However, with imperfect steering, a model can in principle be capable of producing all instances humans may want (i.e. $S_m = T$), and yet produce images $h(m, r)$ with arbitrarily bad reward when steered by a human. Thus, both terms are needed to benchmark model quality under human use.

Many existing ML benchmarks (e.g. image evaluation metrics like Inception Score [70] and Fréchet Inception Distance [32]) only measure producibility. This is partially due to the challenge of measuring steerability, which relies on human reward functions that may be unknown. Recent benchmarks have been proposed for evaluating the success of human-AI interactions [48, 39, 81]. While important for understanding overall efficacy, these benchmarks combine producibility and steerability into a single metric. Decomposing these effects is important for improving models along each dimension, which are optimized with different pipelines; e.g. producibility may be improved through model architecture and training data, while steerability may be enhanced through new interfaces and alignment techniques.

**Benchmark task.** In this paper, we propose and implement a benchmark task for measuring a model's steerability independently of its producibility. This task is revealed in the framework above: measure how well humans can steer a model toward the highest-reward instances it is capable of producing.

Evaluating a model's steerability is challenging because we often don't have access to human reward functions. Even if we did have access to reward functions, evaluating the gap by finding the instance in a model's producible set $S_m$ that maximizes the reward function may be intractable. Here, we describe a simple procedure to induce reward functions that circumvents these challenges. For a given model $m$, we first sample an instance $x_g \sim P$ from a distribution $P$ over the model's producible set $S_m$. We refer to $x_g$ as the *goal instance*. To induce a reward function, we show a human the goal instance $x_g$ and instruct them to use the generative model to reproduce it. Specifically, we instruct them to generate an instance as close as possible to the goal instance $x_g$, as judged by other humans. In this sense, we're inducing an ideal point reward function [20]:

$$r(x) = -d(x_g, x), \tag{3}$$

[2]In practice this function could differ between humans, even if they have the same reward function. For notational simplicity we report a single function $h$, which can be interpreted as the aggregated function over humans.

---

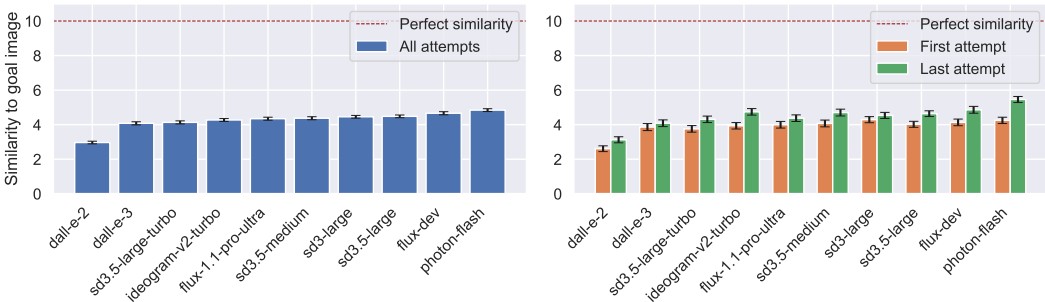

**Figure 2:** Human ratings (0-10 scale) measuring the similarity between human-steered text-to-image model outputs and their corresponding goal images. **Left:** For each model, the average similarity of all user-generated images to their goal images. **Right:** For each model, the average similarity of users' first and last attempted generated images to their goal images. Bars represent single standard errors.

where $d(\cdot, \cdot) \in \mathbb{R}^+$ is the distance function as judged by humans (with the property that $d(x, x) = 0$). Therefore, for a given goal instance $x_g$, the steerability gap is

$$r_{S_m}^* - r(h(m, r)) = \min_{x' \in S_m} d(x_g, x') + d(x_g, h(m, r)) = d(x_g, h(m, r)),$$

where the equality is due to the fact that the goal instance $x_g \in S_m$. Finally, to evaluate the steerability gap, we ask separate human annotators to rate the similarity of $x_g$ and $h(m, r)$. We repeat this process across goal instances and human users to form an aggregate steerability measure for the model $m$.

This task is straightforward and can be applied to any domain where there's a natural notion of human similarity. For example, to implement this benchmark for text-to-image models, we first sample an image from the model's producible set. We then show this image to a human, and instruct them to prompt the model to produce the image (see Figure 1). We then take the image they produce and ask other humans to rate how close the two images are.

## 3 Benchmarking steerability

We now evaluate the steerability of image-generation models using the task described in Section 2. In a large-scale user study, we find that model steerability is poor, both for survey respondents and professional prompt engineers. Overall, annotators rate the attempted reproductions as dissatisfactory 60% of the time. To the extent that there is improvement, more than half of this improvement can be matched by a blind steering mechanism that generates edits of a user's first prompt without knowing the goal image. We find similar results in a smaller user study for steering LLMs (Appendix B).

**Setup.** We use the framework described in Section 2 to study the steerability of text-to-image models. These include models, such as Stable Diffusion [23], that are prompted via text to generate images. We refer to this type of steering as *text steering*. We note that some models allow other mechanisms for steering, such as negative prompts [3] and image inpainting [82]. We focus our study on text prompting because it is the most common mechanism across these models, although we find similar results when we expand to other settings, such as allowing users to experiment with random seeds or control classifier-free guidance settings [34] (see Appendix B).

We study 10 text-to-image models. We consider four variants of the Stable Diffusion models: SD3-large, SD3.5-medium, SD3.5-large, and SD3.5-large-turbo [23]. We also consider two versions of DALL-E: DALL-E 2 [67] and DALL-E 3 [5]. We also consider Flux-dev [6], Flux-1.1-pro-ultra [6], Ideogram-v2-turbo [37], and Photon-flash [57]. We use publicly available APIs for each model: Stability AI for the stable diffusion models, the OpenAI API for the DALL-E models, and the Replicate AI API for all other models.

**Benchmark.** For each model, we sample a goal image from the model's producible set by prompting it with a random image caption from the PixelProse dataset [73]. We then show the goal image to a human user and instruct them to generate an image as close as possible to the goal image. We give them 5 attempts to prompt the model. After each prompt, they are shown the image generated by the model and given the option to refine their prompt to improve the generated image. We repeat this process across goal images and users.

| Model | Imp | POM-1 | POM-5 |
|---|---|---|---|
| DALL-E 2 | 0.60 (0.04) | 0.64 (0.06) | 0.72 (0.05) |
| DALL-E 3 | 0.58 (0.04) | 0.56 (0.07) | 0.52 (0.06) |
| Flux-dev | 0.66 (0.04) | 0.48 (0.07) | 0.64 (0.06) |
| Flux-1.1-pro-ultra | 0.54 (0.05) | 0.49 (0.07) | 0.55 (0.06) |
| Ideogram-v2-turbo | 0.61 (0.04) | 0.46 (0.07) | 0.70 (0.05) |
| Photon-flash | 0.74 (0.04) | 0.45 (0.07) | 0.56 (0.06) |
| SD3-large | 0.51 (0.05) | 0.52 (0.07) | 0.52 (0.06) |
| SD3.5-medium | 0.66 (0.04) | 0.64 (0.06) | 0.68 (0.05) |
| SD3.5-large-turbo | 0.69 (0.04) | 0.56 (0.06) | 0.61 (0.06) |
| SD3.5-large | 0.63 (0.04) | 0.54 (0.07) | 0.68 (0.05) |
| Average | 0.62 (0.04) | 0.54 (0.07) | 0.62 (0.06) |

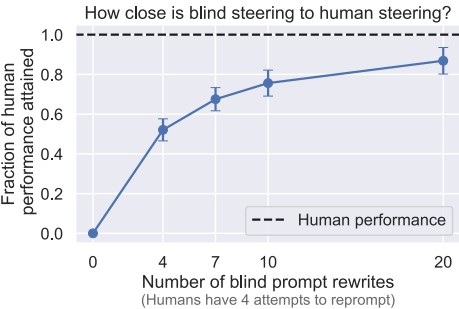

**Figure 3: Left:** Steerability is poor for all models. Imp shows Improvement Rate and POM-1 and POM-5 show Prompt-Output Misalignment on users' 1st and 5th attempts, respectively (standard errors in parentheses). **Right:** When a blind LLM is given the same number of attempts as humans to rewrite a prompt, it achieves more than half of the human improvement. Improvement is measured by the difference between a user's first and best DreamSim score with the goal image (bars represent single standard errors).

Most of the models we consider rely on random seeds to generate images; the same prompt with different random seeds may result in different images. In our surveys, we don't provide users with the random seed used to generate the goal image. This reflects how these models are used in practice: if a model is able to generate an image a user wants, the user typically won't know the specific seed(s) necessary to generate that image. In Appendix B, we repeat our main experiment but allow users to choose random seeds in addition to prompts, finding no significant change in steering performance.

We consider various metrics for judging steering attempts. All metrics use separate human annotators from the ones who perform the steering. *Satisfaction rate* measures whether annotators would be satisfied with the steered image in trying to produce the goal image, according to a four-point scale (very unsatisfied to very satisfied). *Image similarity rating* asks the same question but on a 10-point scale. *Improvement rate (Imp)* is the percent of the time human raters judge the last attempted generated image to be closer to the goal image than the first attempt. *Prompt-Output Misalignment (POM)* measures the percent of the time human judges deem human steering prompts to be better descriptions of the goal images than the corresponding generated images. In other words, POM measures the frequency with which the model's generated output fails to match what a human would reasonably expect from their prompt for recreating a goal image. We compute POM scores for both the 1st prompt (POM-1) and the last prompt (POM-5) that humans attempt. See Figure 20 for a visual summary of these metrics.

We recruit survey participants on the Prolific platform [64]. We received an IRB review and exemption for this study. For all of our surveys, we paid respondents an implied rate of $12.50-$13.50 per hour, and the median survey completion time ranged from 9-15 minutes across tasks. See Appendix D for more survey details. In total, we collect data for 554 goal images, resulting in 2,770 total (goal image, generated image) pairs across 277 different survey-takers. We collect a total of 18,550 ratings across the four metrics. We release all of the data we collect.[3]

**Results.** We find that steerability is poor across models. Annotators rate the attempted reproductions as unsatisfactory 60% of the time; moreover, 27% say they'd be "very unsatisfied" with the attempted reproductions compared to only 10% saying they'd be "very satisfied". The image similarity results on a 10-point scale are depicted in Figure 2. The best model is Photon-flash, while the worst model is DALL-E 2. In general, there is not much difference between different-sized models in the same family, although larger ones perform marginally better. We find an average POM-1 score of 0.54, which means that more than half the time, a human's description aligns with the goal image better than the generated image. See Figure 1 and Figure 11 for examples of steering attempts.

Poor steerability isn't due only to poor initial attempted generations. Figure 3 shows the improvement rate: only 62% of the time are images generated by a human's 5th attempt ranked as more similar to the goal image than images generated by a human's 1st attempt (compared to a baseline of 50%). To assess the impact of longer-term steering experience, we also repeat the main experiment with a small group of professional prompt engineers recruited via Upwork [79]. Across 34 examples, professionals

---

[3]https://github.com/SarahBentley/Steerability

only slightly outperform non-experts, with final similarity scores 10% higher (see Appendix B). However, they also show no reliable improvement over attempts, with even lower improvement rates.

Additionally, we find that even what appears to be improvement can be partially explained by the opportunity for users to try out different prompts. To quantify this, we "blindly" generate edits of a user's first prompt without knowing the goal image by instructing a large language model (LLM) to produce $k$ variations of a user's first prompt. We then measure the maximum similarity score between the $k$ images produced by the blind prompts and the goal image. The results in Figure 3 show that this blind form of steering attains a substantial portion of human improvement; **when humans and a blind LLM have the same number of attempts to rewrite a prompt, the LLM achieves 52% of the human improvement**. Further, 87% of the human improvement can be achieved by a blind LLM with 20 opportunities to rewrite the prompt. These results help calibrate the scale of user improvement by considering "lottery effects" of reprompting. We provide more details in Appendix C.1.

Because our metrics are based on human annotations, they might be subject to human variability. In Appendix B, we repeat our study using model-based similarity metrics; CLIP embedding cosine similarity [66] and DreamSim [25]. We observe patterns consistent with our human annotations (Figure 5).

**Steerability vs. prompt-image alignment.** While there exist metrics for prompt-image alignment like CLIP [31], steerability measures something distinct: user intent. If many images are consistent with a single prompt or if a user has many desiderata that are infeasible to convey in a single prompt, CLIP scores can be high with poor steerability. Still, a natural question is how much of steerability can be attributed to prompt-image (mis)alignment empirically. Figure 6 plots the relationship between each attempt's prompt-image alignment (measured via CLIP) and steerability score. The correlation is low: 0.32. While poor prompt-image alignment is a factor for poor steerability, it cannot fully explain it.

## 4    Comparing producibility and steerability

One way to artificially improve a model's steerability is to limit the outputs it can produce. However, this could come at the expense of producibility, since the model wouldn't be able to generate as many outputs. Here we ask: is there a way to assess whether steerability improvements reflect genuine improvements in steering mechanisms as opposed to differences in producibility?

We first study a setting where there is a controllable tradeoff between producibility and steerability: when steerability is artificially improved by constraining all images to come from the same random seed. Specifically, we consider different versions of Stable Diffusion 3.5 Large Turbo: the default model, along with three versions that constrain the number of random seeds the model can use to produce images. We measure steerability and producibility scores for each version. Figure 7 in Appendix C.1 demonstrates a tradeoff: as the model becomes more constrained, it's easier to steer, but less capable of producing. While there's a tradeoff in this artificial setting, it's possible for a model to have better metrics than another for both producibility and steerability; indeed, while DALL-E 3 is more sophisticated than DALL-E 2, our empirical results show that it does not have worse steerability. Like many other metrics (e.g. Type-I and Type-II errors), it's important to measure multiple dimensions — producibility and steerability — to evaluate and improve models.

We formalize this tradeoff with another decomposition. Consider two image-generation models $M_1$ and $M_2$ with average steerability rewards $R_1$ and $R_2$. Here we expand on the notation in Section 2 by considering each model's distribution over producible images: $p_1(x)$ and $p_2(x)$.[4] Each model also has a steering mechanism that results in some expected reward for each image being reproduced: denote by $\mu_i(x)$ the average reward a user whose goal is to produce image $x$ receives when using $M_i$. The difference in steerability between the two models can be expressed as:

$$R_2 - R_1 = \int p_2(x)[\mu_2(x) - \mu_1(x)]dx + \int p_2(x)\mu_1(x)dx - \int p_1(x)\mu_1(x)dx \qquad (4)$$

$$= \underbrace{\mathbb{E}_{p_2(x)}[\mu_2(X) - \mu_1(X)]}_{\text{improvement due to steering mechanism}} + \underbrace{\mathbb{E}_{p_2(x)}[\mu_1(X)] - \mathbb{E}_{p_1(x)}[\mu_1(X)]}_{\text{improvement due to producible set}}. \qquad (5)$$

The first term holds the producible set constant and measures differences due to the steering mechanism alone. Meanwhile, if the difference is dominated by the second term, improvement

---

[4]Here each $x$ denotes an image, although it can also be characterized as an image and corresponding prompt.

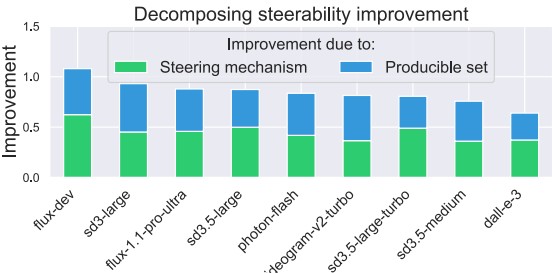

| Model | $R^2$ | MSE |
|---|---|---|
| Baseline | 0.000 (0.000) | 1.061 (0.000) |
| Model only | 0.082 (0.025) | 0.974 (0.071) |
| Model + prompt | 0.459 (0.040) | 0.543 (0.034) |
| Model + image + prompt | 0.493 (0.035) | 0.508 (0.033) |

**Figure 4: Left:** Predictive performance using CLIP-based methods to predict steerability (standard errors in parentheses). **Right:** Each model's steerability improvement over DALL-E 2 broken into two components: one based on differences in the steering mechanism, the other based on differences in the producible set (Equation 5).

| | PixelProse | | Tiles | |
|---|---|---|---|---|
| | Avg. change | % improve | Avg. change | % improve |
| Text steering | 0.025 (0.010) | 54.7% (3.6%) | 0.029 (0.015) | 56.2% (5.1%) |
| Image steering (random proposals) | 0.040 (0.007) | 62.0% (4.9%) | 0.053 (0.005) | 66.8% (2.6%) |
| Image steering (learned proposals) | **0.053 (0.008)** | **74.2% (5.4%)** | **0.072 (0.007)** | **70.7% (3.0%)** |

**Table 1:** Image steering outperforms text steering, with further improvements coming from learning proposal distributions. We show each method's improvement between the first and last generated image, both with the average magnitude of improvement (avg. change), and the percent of time there is an improvement (% improve). Similarity scores are measured with DreamSim [25]. Standard errors are in parentheses.

cannot be attributed to differences in the steering mechanism; only the producible set is changing. While Equation 5 isn't a causal quantity, it accounts for how differences are distributed, and it's mathematically equivalent to decompositions of group differences used to study economic outcomes [24]. See Appendix A for a derivation.

Estimating the terms in Equation 5 involves training a predictor $\hat{\mu}_i(X)$ to predict steerability from inputs for each image-generation model. How well can ML methods predict steerability? We consider four predictive models: a baseline using only the data mean, a model using only the model being steered as a feature, a model also using a user's prompt, and a model using both the goal image and the user's prompt. We split the data collected in Section 3 into 80/20 train/test splits. To predict steerability, we encode the prompts and/or images with CLIP and fine-tune CLIP layers and a regression head to predict DreamSim scores from the resulting embeddings and a one-hot encoding of the model name. The training and analyses were performed on a single A100 GPU.

Figure 4 shows the predictive performance of each model on held-out data. We find that the highest-performing model can explain nearly half of the variance in steerability scores. We imagine that more sophisticated prediction methods can do even better.

We now estimate Equation 5 using Monte-Carlo sampling. Figure 4 reports results using the best predictive model. We compare each model to DALL-E 2, since it is the only model for which other models have consistently better steerability. Across models, half of this improvement can be attributed to genuine improvements in steering mechanisms, while the other half is attributed to differences in producibility sets. This means, for example, that the images that DALL-E 3 is more likely to produce than DALL-E 2 are easier for people to steer to regardless of the model.

## 5  Steerability can be improved

The results in Section 3 suggest the need to focus on improving the steerability of generative models. Here, we consider a simple alternative form of steering image models, based on selecting images instead of refining prompts. In a user study with 500 human steerers, this new steering mechanism results in more than 2x improvements over text steering.

**Image steering.** Our results in Section 3 suggest that prompting is an inefficient steering mechanism. Here, we consider a simple alternative form of steering, which we refer to as *image steering*. As before, humans begin by providing a prompt to a model, which is then used to generate an image. However, instead of relying on humans to rearticulate a new prompt, the model returns variations of the image. The user can either select a preferred variation or stick with their original image, at which point the model suggests new variations. This procedure does not require rearticulating textual prompts; instead, the model does the "rearticulating" while the human chooses between options.

Specifically, given a current image $x$, new images $x'$ are sampled from a *steering distribution* $q(x'|x)$ and suggested to the user. The efficacy of image steering depends on this steering distribution. Which image variations should be suggested? We consider two approaches: one is to suggest random perturbations in latent space. However, some sets of image variations will be more helpful to humans than others, regardless of the goal image they're trying to generate. So we also consider a second approach, which learns a steering distribution that suggests a set of images likely to improve steering (independently of the goal image). We use a simple RL technique that learns the optimal distance between current and suggested images by simulating human steerers offline; we describe more details in Appendix C.2.

**Results.** We use our benchmark to evaluate image steering. We focus on two sampling distributions to generate goal images. We first consider prompts from the PixelProse dataset to generate goal images (as in Section 3). To also understand the effects in a domain where human articulation is more difficult, we consider images generated from prompts about abstract geometric patterns. Specifically, we prompt Claude-3.5-Sonnet to provide 25 variations of prompts involving geometric patterns on tiles and use these prompts to generate images. For the image steering methods, we only suggest two new images per round and do not allow users to update their text prompt. We also limit the number of attempts to 5 to allow for a direct comparison to text steering.

Because our goal is to compare steering methods rather than generative models, we use a fixed model: Stable Diffusion 1.4 [68], which is open-source, allowing us to perform perturbations. Moreover, at 860M parameters, it is large enough to produce high-quality images but small enough to enable efficient image generation given our academic-level computing constraints. We recruited 500 participants from Prolific to perform steering. Because image steering involves perturbing the model's latent space, we could not rely on black-box APIs. Instead, we used a server of 8 H100 GPUs to perform image generation and perturbation; we found the process to be efficient. We use DreamSim to rate the similarity of the generated and target images. Because all methods use prompting to generate the first image, we evaluate methods by their improvement between the first and last generated image.

The results are summarized in Table 1. Across both domains and evaluation metrics, image steering outperforms text steering. While learning the proposal distribution expands on these improvements, there is already ample improvement without it. In both domains, the average total improvement is more than 2x the improvement for text steering. Figure 9 shows how generated images improve as human users continue to steer. While the improvement for text steering is relatively flat, image steering has steeper improvements (see Figure 12 and Figure 13 for examples). These results demonstrate that good steering performance on our benchmark is attainable; the fact that this simple image steering mechanism attains good results suggests that more sophisticated procedures can do even better.

# 6 Related work

Our work on evaluating steerability is contextualized by a large body of existing work focused on evaluating and improving human-AI interaction. For example, Lee et al. [48] develop a framework for evaluating human-language model interaction, and further research explores gaps in current human-AI interactions and proposes methods for improvement [13, 80, 44, 14, 56, 38, 10]. In the context of text-to-image generation models, many benchmarks involving steerability evaluate the process of *editing* images with generative models [82, 76, 4, 72, 26]. Editing is a useful strategy for steering so these benchmarks clearly measure steerability. However, unlike our framework, they are not measuring steerability in isolation — in order to edit a generated image to perfectly match a goal image, a model must be capable of producing the goal image in the first place.

In the LLM literature, steerability and producibility are also typically evaluated in conjunction [11, 63, 58, 51]. For example, Zamfirescu-Pereira et al. [85] conduct a user study finding that non-AI experts struggle to make systematic progress in prompt design with chatbots. While these evaluation frameworks are important for understanding overall efficacy, they combine producibility

and steerability into a single metric. Decomposing these effects is crucial for improving models along each dimension, and our framework studies steerability in isolation.

Our main study focuses on evaluating the steerability of image generation models. There exist many text-to-image generation benchmarks for measuring the quality of generated images in non-interactive experiments [54, 62, 28, 69, 35], some of which have found systematic failures (e.g. due to shared CLIP encodings) [78]. Other evaluation metrics include measures of: (i) text-image alignment [31, 49, 50]; (ii) similarity to a reference image (often using embeddings from models like CLIP) [87, 83, 66, 9]; (iii) image quality [71]; and (iv) human preferences [25, 43, 84, 65, 47]. Most frequently, these metrics are used to measure producibility and *not* steerability, as is the goal of our paper.

In this paper we choose to evaluate text steering and image steering as our primary mechanisms. However, prior work highlights many alternative steering mechanisms. Examples in image generation include negative prompting [3], manipulating prompts and their embeddings [17, 29], adding conditional controls [86], using human image edits [88], and generally leveraging textual and image inputs to guide the model [61, 8, 30]. Examples for other generative tasks include extracting and moving along interpretable axes in the latent space [21, 7, 55, 75, 16], updating model weights [18, 15, 42], and editing prompts [41, 45]. These mechanisms are all amenable to our framework and test; while we find that image steering outperforms text steering in our surveys, we do not attempt to suggest it is the best overall steering mechanism. Future work should explore these other approaches.

Similar to how we learn steering distributions for image steering, existing approaches use reinforcement learning to align models with human preferences [40]. For example, Hilgard et al. [33] develop a training procedure to optimize machine representations for human-model collaboration. Reinforcement learning from human feedback (RLHF) is also used to improve the alignment of LLMs [12, 2, 52] or generated images [47, 53] with human preferences. While these methods are similar to our technique in that they use human preferences to adjust model behavior, their goals are different; RLHF is designed to align model outputs with human preferences, while our technique is designed to align model *steering* with human intuitions.

Our work complements recent findings by Jahani et al. [39] and Vodrahalli & Zou [81], who conduct large-scale experiments on human uses of generative models. Like our work, these experiments focus on how well humans can prompt image-generation models to produce certain goal images. A key difference is that we focus on measuring steerability in isolation from producibility, so we only include reference images in a model's producible set.

# 7 Conclusion

This paper introduced a mathematical framework and benchmark for evaluating the steerability of generative models. We implemented the benchmark, which revealed shortcomings in text-to-image models. We also showed steerability can be improved, achieving progress with simple alternative mechanisms.

Our work has several limitations and future directions worth noting. While there are many methods for steering models, we focused on text steering and image steering. However our framework can accommodate many forms of steering, and future work should explore other approaches. Additionally, we only considered a basic implementation for suggesting images that lead to better steering. The fact that even this basic implementation achieved improvements suggests that more sophisticated methods applied to larger models could yield even greater benefits. Finally, while we've focused on whether steerability improves during single interactions, it would be interesting to study longer time-horizons, e.g. over the course of months.

We designed the benchmark with the goal of avoiding copyright issues: each image used in the benchmark is generated by a text-to-image model, and is not drawn from any external, copyrighted dataset. Still, as with any use of text-to-image models, it's possible that a model may produce a copyrighted image. Moreover, improved steerability could also facilitate malicious uses. However, understanding and quantifying steerability is an important prerequisite for responsible deployment and mitigation planning.

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

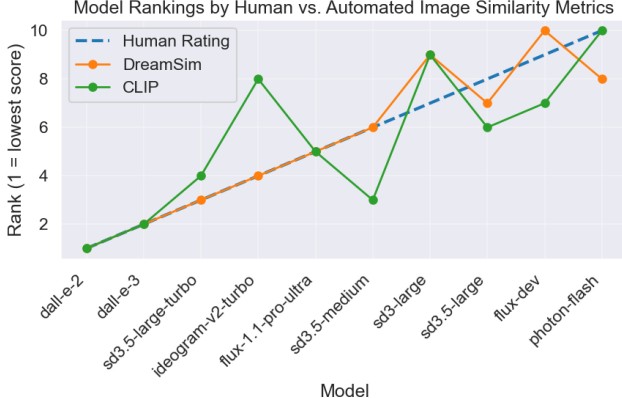

**Figure 5:** Rankings of the steerability of image generation models according to human ratings, DreamSim, and CLIP embedding cosine similarity.

# A    Derivation of Equation 5.

Here we derive Equation 5. Consider two models, $m_1$ and $m_2$, and denote each model's distribution over producible instances: $p_1(x)$ and $p_2(x)$. Each model also has a steering mechanism that results in some expected reward for each image being reproduced: denote by $\mu_1(x)$ the average reward a user whose goal is to produce image $x$ receives from model 1, with $\mu_2(x)$ the analogous quantity for model 2. Formally, $\mu_i(x) = \mathbb{E}[r_X(h(M, r_X))|X = x, M = m_i]$, where the reward function is subscripted by the goal instance $x$ to make its dependence explicit. To simplify notation, we will define $S_m(x) = r_x(h(m, r_x))$, so that $\mu_i(x) = \mathbb{E}[S_M(x)|M = m_i]$.

Denote by $R_1$ the average steerability for model 1 and $R_2$ the analogous quantity for model 2. By the law of iterated expectation,

$$R_i = \mathbb{E}[S_M(X)|M = m_i] = \mathbb{E}[\mathbb{E}[S_M(X)|X = x, M = m_i]|M = m_i] = \mathbb{E}_{p_i(x)}[\mu_i(X)]. \quad (6)$$

Therefore, we can write

$$R_2 - R_1 = \mathbb{E}_{p_2(x)}[\mu_2(X)] - \mathbb{E}_{p_1(x)}[\mu_1(X)]$$
$$= \int p_2(x)\mu_2(x)dx - \int p_1(x)\mu_1(x)dx.$$

Add and subtract the "cross-term" $\int p_2(x)\mu_1(x)dx$:

$$R_2 - R_1 = \int p_2(x)\mu_2(x)dx - \int p_2(x)\mu_1(x)dx + \int p_2(x)\mu_1(x)dx - \int p_1(x)\mu_1(x)dx$$
$$= \int p_2(x)[\mu_2(x) - \mu_1(x)]dx + \int p_2(x)\mu_1(x)dx - \int p_1(x)\mu_1(x)dx$$
$$= \underbrace{\mathbb{E}_{p_2(x)}[\mu_2(X) - \mu_1(X)]}_{\text{improvement due to steering mechanism}} + \underbrace{\mathbb{E}_{p_2(x)}[\mu_1(X)] - \mathbb{E}_{p_1(x)}[\mu_1(X)]}_{\text{improvement due to producible set}}.$$

# B    Additional results

In this section we present additional analyses of our steerability benchmarks for text-to-image models and LLMs.

## B.1    Comparing human-judged metrics to automatic metrics.

For our main experiment, we use human ratings to evaluate model similarity. In Table 2 we consider algorithmic similarity metrics: DreamSim [25] and CLIP [31]. As shown in Figure 5, we find that the DreamSim and CLIP image similarity metrics are relatively aligned with human ratings.

**Table 2:** The DreamSim scores and CLIP embedding cosine similarities of generated and goal images from the benchmark on steering text-to-image generative models. DreamSim Avg and CLIP Avg describe the average similarity of each model's generated images with their corresponding goal images.

| Model | DreamSim Avg | CLIP Avg |
|---|---|---|
| DALL-E-2 | 0.52 (0.01) | 0.75 (0.01) |
| DALL-E-3 | 0.62 (0.01) | 0.78 (0.01) |
| Flux-dev | 0.70 (0.01) | 0.83 (0.01) |
| Flux-1.1-pro-ultra | 0.66 (0.01) | 0.82 (0.01) |
| Ideogram-v2-turbo | 0.66 (0.01) | 0.83 (0.01) |
| Photon-flash | 0.68 (0.01) | 0.85 (0.01) |
| SD3-large | 0.68 (0.01) | 0.84 (0.01) |
| SD3.5-medium | 0.67 (0.01) | 0.81 (0.01) |
| SD3.5-large-turbo | 0.65 (0.01) | 0.82 (0.01) |
| SD3.5-large | 0.67 (0.01) | 0.82 (0.01) |
| Average | 0.65 (0.00) | 0.82 (0.00) |

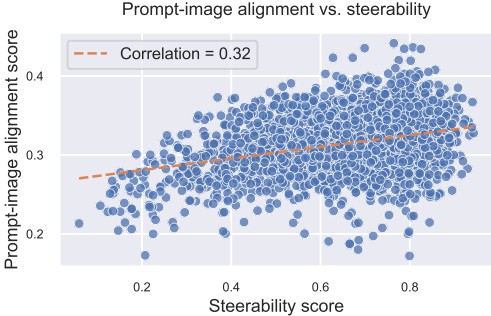

**Figure 6:** Steerability scores are only weakly correlated with prompt-image alignment scores based on CLIP. We calculate CLIPScore by computing the cosine similarity between normalized CLIP embeddings of the prompt and image. While cosine similarity is on a scale from −1 to 1, we find most scores are between 0 and 0.4, aligning with prior work [31].

## B.2   LLM steerability.

We also apply our framework to evaluate the steerability of large language models (LLMs). Just as in image generation, we provide users a goal instance — a piece of text — and instruct them to iteratively prompt an LLM to output the goal text. We prohibit users from using words in the goal text to prevent them from simply prompting the LLM to repeat it. We study 5 LLMs: Gemini-1.5-flash [77], Gemini-2.0-flash-exp [27], GPT-4o [36], Claude-3.5-Sonnet [1], and Llama-3.3-70B-Instruct-Turbo [22].

For each LLM, we sample a goal text from its producible set by prompting it to rewrite a news headline from the Kaggle News Category dataset using a particular style and tone [59, 60]. We provide users with both the original and goal headlines, and instruct them to prompt the LLM to rewrite the original headline in order to output the goal headline. Importantly, users are allowed to use words from the original but not the goal headline in their prompt, so they cannot directly instruct the LLM to output the goal headline. We give users 5 attempts to prompt the model. See Table 4 for an example. More survey details are in Appendix D.

We use two kinds of human annotations to judge the steerability of LLMs: satisfaction rate (a 4-point scale from "very unsatisfied" to "very satisfied") and improvement rate (as before). We recruit survey participants on Prolific, yielding 132 respondents, 118 goal headlines, and 2,030 total ratings. We again find poor steerability: human raters are very satisfied with the generated headline only 17% of the time. Additionally, the average improvement rate includes 50% in its margin of error, indicating that improvements over steering attempts occur only half the time.

**Table 3:** Few users are very satisfied with the generated headlines produced from steering LLMs, and human raters find improvements between the first and last generated headline about half the time. "Satisfaction Rating" shows the average satisfaction of raters on a scale from 1 (very unsatisfied) to 4 (very satisfied). "Very Satisfied" describes the fraction of satisfaction ratings equal to 4, and "Improvement" measures the fraction of time a user's last headline is rated as more similar than their first.

| Model | Satisfaction Rating | Very Satisfied | Improvement |
|---|---|---|---|
| Claude-3-5-Sonnet-20241022 | 2.54 (0.06) | 0.15 (0.02) | 0.59 (0.06) |
| Gemini-1.5-Flash | 2.59 (0.07) | 0.21 (0.03) | 0.45 (0.06) |
| Gemini-2.0-Flash-Exp | 2.48 (0.07) | 0.17 (0.03) | 0.44 (0.05) |
| GPT-4o | 2.65 (0.07) | 0.17 (0.03) | 0.40 (0.06) |
| Llama-3.3-70B-Instruct-Turbo | 2.48 (0.08) | 0.19 (0.03) | 0.42 (0.06) |
| Overall | 2.54 (0.03) | 0.18 (0.38) | 0.47 (0.03) |

**Table 4:** In our benchmark on the steerability of LLMs, humans are instructed to prompt an LLM to rewrite the original headline as the goal headline, without using words that appear in the goal headline but not the original. Below we show five rounds of user instructions and the resulting model outputs.

| Item | Text | |
|---|---|---|
| Original Headline | What James Gandolfini's Son Learned Last Night | |
| Goal Headline | Progeny of 'Sopranos' Icon: A Critical Analysis of Michael Gandolfini's Artistic Evolution | |
| Prompt Preamble | I will give you instructions to rewrite this news headline: "What James Gandolfini's Son Learned Last Night". Only return the rewritten headline. Here are my instructions: | |
| **Attempt** | **Instructions (User Input)** | **Generated Headline (Model Output)** |
| 1 | Try to make it sound more like a scientific article, looking into the matter with a dose of skepticism; make it sound like the artist's creation is novel, as if it collected from the previous creations and presented something more. | Recent Analysis Suggests Novel Patterns in Gandolfini Heir's Artistic Evolution: A Critical Examination |
| 2 | *Same as above, then:* Now, for the first part of the title, do it in a more artsy, creative way; imply that the artist represents a class of itself; imply a parent-child relationship with art; highlight the high-pitched voice of the artist. | High Notes of Heredity: A Scientific Investigation into the Next-Generation Gandolfini's Novel Artistic Synthesis |
| 3 | *Same as above, then:* Now make it less scientific and more like a journal title. | The High Notes of a Legacy: How Gandolfini's Heir Reimagines the Family Opus |
| 4 | *Same as above, then:* Now use a bit less fancy words after: say the artist's name and surname after. | The Sopranos' Echo: How a Son's High Notes Carry His Father's Legacy |
| 5 | *Same as above, then:* Now use less fancy words; include the artist's full name at the end. | A New Voice in Artistic Lineage: Michael Gandolfini Explores Depths of Character |

In Table 3 we show complete results from this experiment. Table 4 contains examples of headlines.

### B.3 Allowing users to choose seeds during text steering.

To evaluate text steering in its most flexible form, we conduct an additional survey in which we allow users to provide seeds in addition to text prompts when generating images. Thus if a user happens to find a seed with generated images that have a similar style to the goal image, they can continue using that seed. If not, they can try different seeds. In contrast, in our prior text steering survey we did not specify seeds, so each iteration of image generation used a random seed. The seed-choosing survey includes all image generation models in our study except DALL-E 2 and DALL-E 3, which do not allow the specification of seeds in their API. We find that allowing users to specify the seed results in no statistically significant changes in their average DreamSim scores. We show the results in Table 5.

### B.4 Steering with classifier-free guidance.

To evaluate text steering with a more sophisticated steering mechanism, we conduct an additional survey allowing users to specify classifier-free guidance (CFG) settings to Stable Diffusion 3.5 Medium. Classifier-free guidance is a technique used in generative models, where the model generates samples conditioned on both the input and a guidance signal, without relying on an explicit classifier, to improve the alignment of generated outputs with user intentions [34]. We focus specifically on

**Table 5:** The average DreamSim score per attempt when users can vs. cannot choose their own seed during image generation. We find no statistically significant change in the similarity of generated images to goal images, as measured by DreamSim.

| Attempt | Choosing Seed | Not Choosing Seed |
|---|---|---|
| 1 | 0.663 (0.017) | 0.639 (0.008) |
| 2 | 0.671 (0.017) | 0.661 (0.007) |
| 3 | 0.665 (0.019) | 0.683 (0.007) |
| 4 | 0.682 (0.018) | 0.684 (0.007) |
| 5 | 0.690 (0.017) | 0.695 (0.007) |
| Average | 0.674 (0.008) | 0.672 (0.003) |

Stable Diffusion because it is one of the few that allows CFG inputs in its API. In the survey, we allow users to choose a number from 1 to 10 for the guidance scale. We instruct them to choose how strictly they want the image generation process to adhere to their prompt text, with higher values keeping their images closer to their prompts, but potentially resulting in unrealistic images. We recommend users a default value of 4. Using this survey, we can compare the steerability of Stable Diffusion 3.5 Medium with and without CFG. Our results are shown in Table 6. We find no statistically significant improvement in steering when users can specify CFG parameters.

**Table 6:** Comparison of average steering performance with and without CFG across 5 attempts, as measured by DreamSim.

| Attempt | Without CFG | With CFG |
|---|---|---|
| 1 | 0.638 (0.020) | 0.664 (0.018) |
| 2 | 0.662 (0.020) | 0.677 (0.017) |
| 3 | 0.676 (0.015) | 0.667 (0.017) |
| 4 | 0.681 (0.015) | 0.703 (0.015) |
| 5 | 0.693 (0.015) | 0.682 (0.017) |
| **Average** | 0.670 (0.008) | 0.679 (0.007) |

### B.5 Steering performance of prompt engineers.

How much does training or experience in steering generative models improve steerability? To answer this question, we compare the steering performance on non-experts and prompt engineers. Our main experiment was conducted by non-experts on the Prolific platform, so we repeat our experiment with 5 prompt engineers recruited on the Upwork platform [79]. To reduce noise from different models and while having a direct comparison between the user groups, we restrict steering to Stable Diffusion 3.5 Large Turbo. Our results are shown in Table 7. Across 34 steering rounds we found prompt engineers were better than the average steerer, but only slightly: final similarity scores were only 10% higher than for non-experts. Moreover, we find that professional prompt engineers don't reliably improve between their first and last attempts, with even lower improvement rates than among the non-expert cohort.

## C Additional details

### C.1 Evaluating blind steering.

To evaluate the extent to which users' steering improvement rates can be attributed to learning (gaining an understanding of how to steer the model) vs. random chance, we prompt an LLM to blindly

**Table 7:** Per-attempt DreamSim scores for regular users and prompt engineers over five attempts.

| Attempt | Regular users | Prompt engineers |
|---|---|---|
| 1 | 0.631 (0.019) | 0.767 (0.014) |
| 2 | 0.636 (0.020) | 0.772 (0.010) |
| 3 | 0.669 (0.018) | 0.777 (0.014) |
| 4 | 0.654 (0.019) | 0.772 (0.014) |
| 5 | 0.670 (0.016) | 0.741 (0.025) |
| **Average** | 0.652 (0.008) | 0.766 (0.007) |

steer image generation models. Specifically, we randomly sample a set of 100 human steering traces from our prior experiment, each of which contains a goal image and five prompts and generated images, corresponding to the human's 5 steering attempts. To have an LLM blindly perform steering, we prompt GPT-4o to produce variations of the user's first attempt prompt for all 100 goal images, then use these variations to generate images. Importantly, we do not provide GPT-4o any additional information about the goal image. For each of the generated images, we calculate the DreamSim similarity to the goal image to evaluate the effectiveness of blind steering.

We repeat this experiment four times, prompting GPT-4o to produce 4, 7, 10, and 20 variations of the first attempt prompts. For example, to produce 4 variations of a user's first attempt prompt, we prompt GPT-4o as follows: "*Provide four different, more detailed variations of this description and label them (1), ..., (4): ...*". We calculate the maximum DreamSim score (measuring the similarity of the generated images to the goal image) among the LLM's prompt variations and calculate how much it improves over the user's first attempt DreamSim score. For each goal image, we consider the LLM's improvement score to be the difference between its highest DreamSim score and the human's first attempt DreamSim score. If this difference is negative, meaning the LLM's best score is worse than the human's first attempt, the improvement is 0. Likewise, we consider the human's improvement score to be the difference between their best score and their first attempt score, according to DreamSim. To calculate the expected human improvement due to random chance, we divide the mean LLM improvement score by the mean human improvement score. Thus this quantity summarizes the percentage of human steering progress that can be achieved through arbitrarily varying their first prompt. The results are shown in Figure 3.

**Artificially improving steerability hurts producibility.** Constraining all images to be generated by the same random seed might artificially improve steerability while worsening producibility since the model wouldn't be able to generate as many outputs. To study the relationship between model steerability and producibility in Figure 7, we considered different versions of Stable Diffusion 3.5 Large Turbo: the default model, along with three versions that constrain the number of random seeds the model can use to produce images. To compute steerability, we ran the survey described in Section 3 but constrained the goal image and all user-generated images to be in the relevant set of random seeds. When we constrain to one seed, this means that each image is generated using the same random seed as the goal image. When we constrain to two seeds, this means that the goal image is generated using a single random seed (e.g. 42), and each attempt the user has to generate the goal image either uses that same random seed (42) or a different, fixed random seed (e.g. 43); for each attempt, this choice is sampled uniformly at random. We consider 1, 2, 3, and the default set of random seeds (4294967294). We measure steerability using the DreamSim similarity score between goal and generated images, averaging over rounds and attempts.

In contrast, producibility here refers to how similar the closest image in a model's set of producible images comes to some goal image. To measure this, we first sample a prompt uniformly at random from PixelProse [73], which we then use to generate an image from a non-Stable Diffusion model. We then ask: how close can the Stable Diffusion model under consideration come to generating this image? Intuitively, models that are able to produce more random seeds can produce a larger set of images and could come closer. However, finding the optimal image in the model's producible set is an optimization challenge, because it requires a combinatorial optimization over possible prompts. Instead, we approximate this optimization by searching over random variations of the prompt used

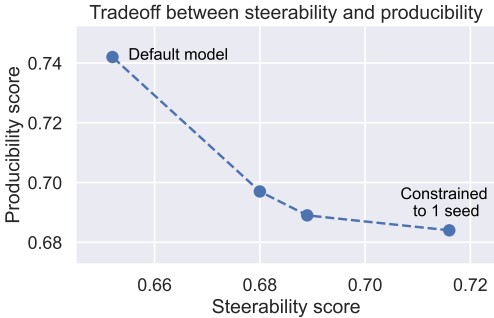

**Figure 7:** While artificially decreasing the number of images a model can produce improves steerability, it results in worse producibility. Each point represents Stable Diffusion 3.5 Large Turbo constrained to a different number of seeds. Similarity scores are calculated with DreamSim [25].

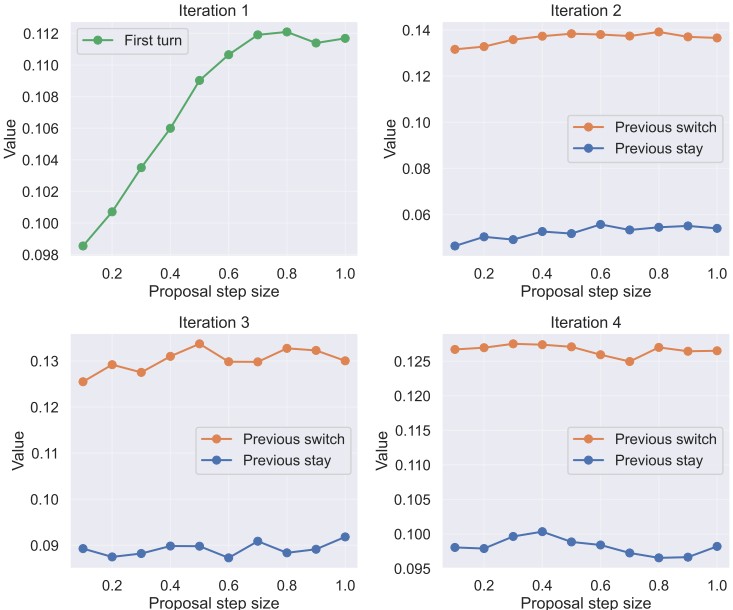

**Figure 8:** The value function for each round of image generation.

to generate the original image. Specifically, we prompt an LLM to generate slight variations of the original prompt used to generate the image (using the same method as in Appendix B), and then generate images for each prompt variation and each random seed. We then find the generated image that is closest to the original image using DreamSim, which we report as our similarity score. When there are more than 30 possible (prompt, seed) variations, we subsample to only include 30 to make the problem tractable. Overall, we perform this procedure for 50 different goal images for each of the 4 Stable Diffusion variations under consideration. The results are depicted in Figure 7.

## C.2 Reinforcement learning.

In this section, we provide implementation details for our reinforcement learning technique for suggesting images to users.

The efficacy of image steering depends on the steering distribution $q(x'|x)$. How should variations of images be suggested to human users? The steering distribution depends on the generative model being used, and here we consider models like diffusion models [74] that are based on transformations of noise vectors and text embeddings. That is, denote an image $x = f(z)$, where $z \in \mathbb{R}^D$ is a vector of random noise concatenated with a prompt embedding. One possible steering distribution is to

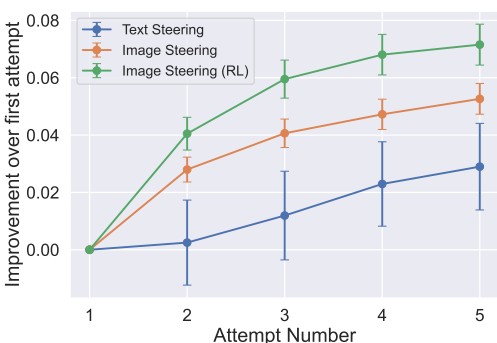

**Figure 9:** Improvement with image steering is more than twice that of text steering on the tiles dataset. Similarity is measured with DreamSim. Single standard errors in parentheses.

sample $\epsilon \sim \mathcal{N}(\mu, \sigma^2)$ with $\epsilon \in \mathbb{R}^D$ and then to decode a new image $x' = f(z + \epsilon)$. This is the *random sampling* mechanism we consider.

However, some sets of suggested image variations will be more helpful to humans than others, regardless of the goal image they're trying to generate. For example, consider two sets of suggested images, one of which contains images that humans find similar and the other that contains images that humans find distinct from one another yet are all similar to the current image. The variety of the latter set makes it more likely that a human can reach their goal image.

Instead of using random sampling as a steering distribution, we propose a reinforcement learning (RL) technique that learns a steering distribution in order to maximize human steering capabilities. Specifically, we parameterize the steering distribution $q_\phi(x'|x)$ with parameters $\phi$. Slightly modifying notation from Section 2 to make the steering function's dependence on $q_\phi$ explicit, denote by $h(m, r|q_\phi) \in S_m$ the instance that is produced when a human with reward function $r$ interacts with a model $m$ with steering distribution $q_\phi$. The goal is to optimize

$$\arg \max_\phi \mathbb{E}_r \left[ r(h(m, r|q_\phi)) \right]. \tag{7}$$

We consider learning a simple steering distribution that suggests image variations as a function of the human steerer's behavior. We consider learning the optimal size of the perturbation. Specifically, we define the steering distribution as $q(x'|x, t, s)$, where $t \in \mathbb{N}$ is the attempt number and $s \in \{0, 1\}$ indicates whether the user stayed with the previous image suggestion or chose a new one. Rather than perturbing both the text embeddings and latent representations, we found that focusing solely on the latent space was sufficient. For a given latent vector z, we generate variations using a mixture of the original latent and a random noise vector:

$$z' = \sqrt{\frac{1}{s^2 + (1-s)^2}} \left( (1-s)z + s\epsilon \right) \tag{8}$$

where $s \in [0, 1]$ is the mixture scale and $\epsilon \sim \mathcal{N}(0, I)$ is standard Gaussian noise. The normalization factor ensures the variance of z' matches that of z. This perturbation scheme preserves the model's learned manifold better than additive noise while allowing controlled exploration.

In principle, the policy $\phi$ can be optimized with humans-in-the-loop, e.g. by measuring and optimizing the reward for different steering distributions when humans use them. However, this can be expensive as it requires many human users. Instead, we simulate a human's behavior with an agent; given a human's prompt for a goal image (collected offline), the agent always chooses the image suggestion it deems most similar to the goal image, as measured by the DreamSim similarity metric [25]. The reward for a single episode is then the similarity between the final image and the goal image. Importantly, while we have access to the goal image at train time (e.g. to choose the image that's most similar), in order to mimic real world use, the policy *does not* depend on the goal image.

We implement this procedure by discretizing the continuous mixture scale space into 10 equally spaced buckets between 0.1 and 1.0. The policy must choose a mixture scale for each of the 4 rounds of interaction. To handle the combinatorial nature of this sequential decision problem, we decompose

the policy into independent decisions per round. We use Thompson Sampling to balance exploration and exploitation. For each (round, bucket) pair, we maintain empirical reward statistics (counts and means). The posterior for each pair is approximated as a normal distribution:

$$\mu_{r,b} \sim \mathcal{N}\left(\hat{\mu}_{r,b}, \frac{\sigma^2}{n_{r,b} + 1}\right) \tag{9}$$

where $\hat{\mu}_{r,b}$ is the empirical mean reward, $n_{r,b}$ is the number of times bucket b was chosen in round r, and $\sigma^2$ is a prior variance parameter (set to 1.0 in our experiments).

During training, we:

1. Sample a mean from each bucket's posterior for each round
2. Choose the bucket with highest sampled mean
3. Play an episode using the corresponding mixture scales
4. Update statistics for chosen (round, bucket) pairs using the episode reward

The reward for an episode is the improvement in similarity between the final generated image and the goal image, compared to the initial generated image.

We trained for 60,000 episodes using Stable Diffusion 1.4 as the base model. Each episode used a different prompt and reference image sampled from our dataset. Episodes were run with 4 rounds of interaction and 2 image variations per round. We used the DreamSim perceptual similarity metric both for the simulator's choices during training and for evaluation.

## D  Survey details

We recruited survey participants on the Prolific platform [64]. For all of our surveys, we paid respondents an implied rate of $12.50-$13.50 per hour, and the median survey completion time ranged from 9-15 minutes across tasks. We recruited different users for each survey. We received an IRB review and exemption for this study. We conducted the following surveys on text steering for image generation models: steering (Figure 14, Figure 15), generated vs. goal image similarity ratings on a 10-point scale (Figure 16), improvement ratings (Figure 18), generated image satisfaction ratings on a 4-point scale (Figure 17), prompt-output misalignment ratings (Figure 19), steering with allowing users to choose seeds, and steering with a constrained number of seeds. We conducted three analogous surveys on text steering for LLMs: steering, generated vs. goal headline satisfaction ratings on a 4-point scale, and improvement ratings. For all experiments involving rewriting multiple prompts, we always used rewrites from scratch. That is, a user sees their previous prompt, they're given the opportunity to edit it, and then submit it to the model. Finally, we conducted surveys for image steering with and without RL, and in both the general and tiles domains. See Figure 14 and Figure 15 for examples.

Since it can be difficult to rate the similarity of images, we calibrate users in the similarity surveys by (1) consecutively showing them 5 generated images for the same goal image (in a random order) and (2) providing them at least one duplicate (goal image, goal image) pair for each of the 6 goal images they see (in a random order). Providing users duplicates gives them a chance to see how a rating of 10 should look, and ordering goal images consecutively allows users to go back and forth between images to be more consistent in their ratings.

## E  Additional examples

Table 4 shows a user's attempt at steering an LLM toward rewriting a headline as the corresponding goal headline. Meanwhile, Figure 12 shows examples of users performing image steering. Additionally, Figure 13 shows examples of users leveraging image steering specifically to reproduce goal images of tile patterns.

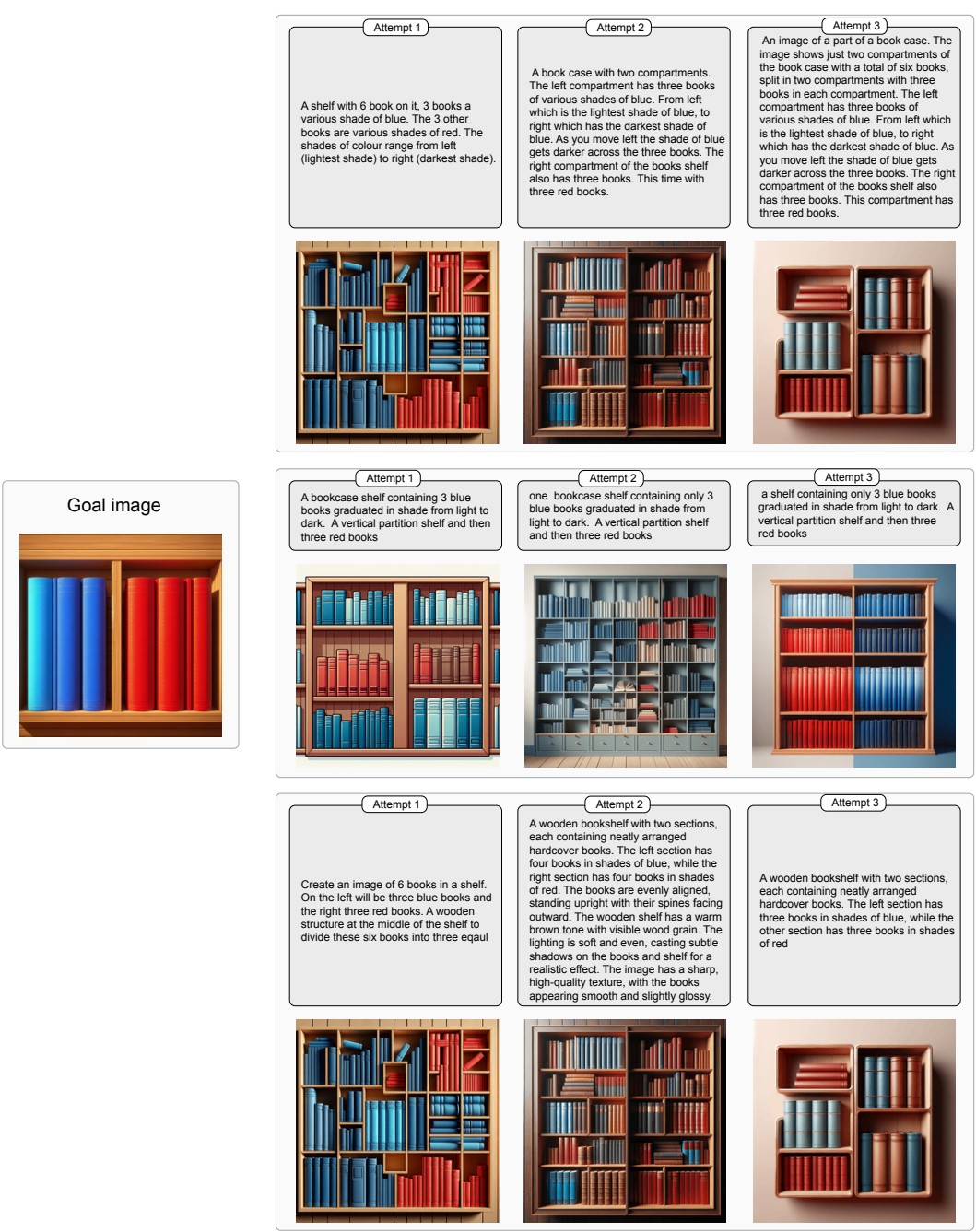

Figure 10: Full prompts for the examples in Figure 1

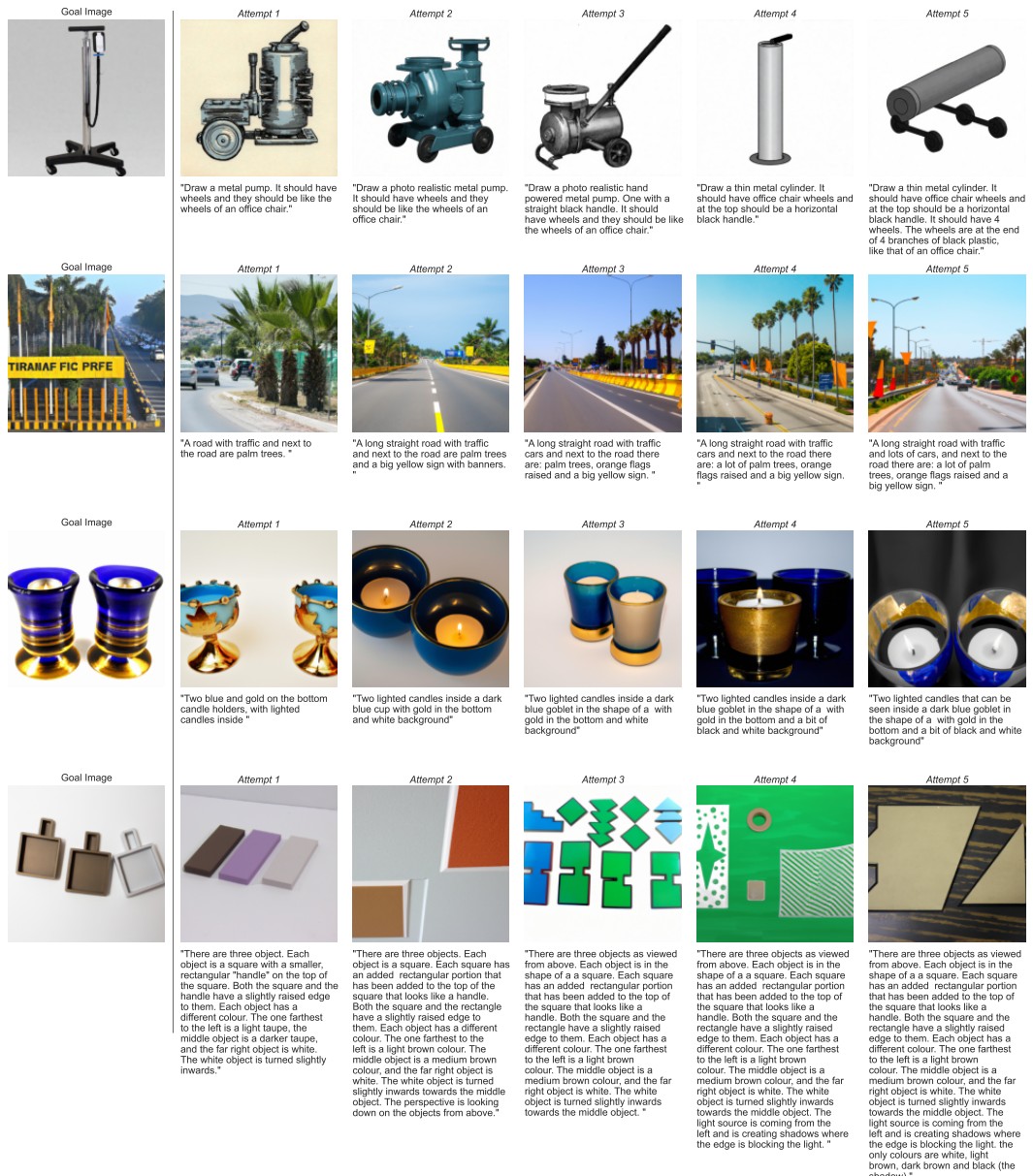

**Figure 11:** Examples of steering text-to-image generative models using text.

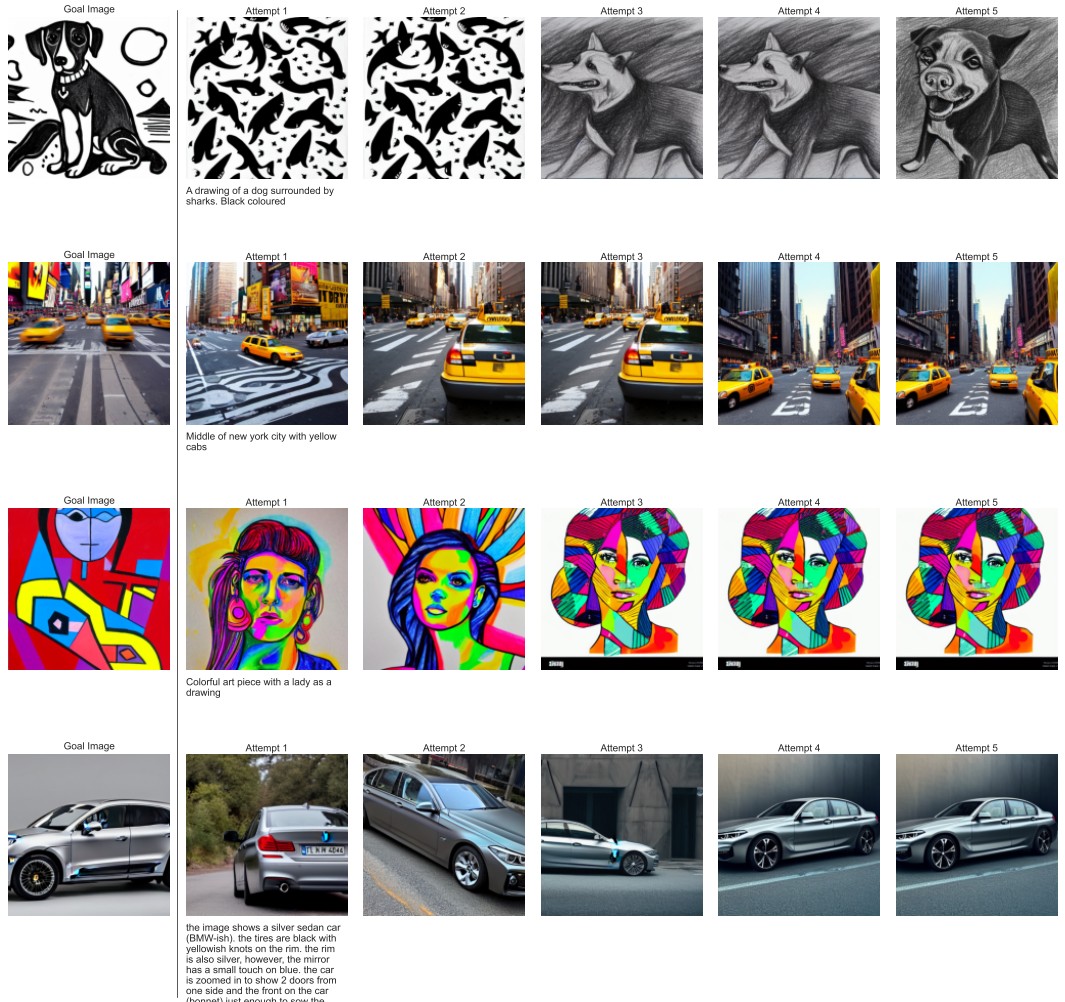

**Figure 12:** Examples of steering image generation models with image steering; at each iteration, humans have the choice to accept or reject a suggested edit.

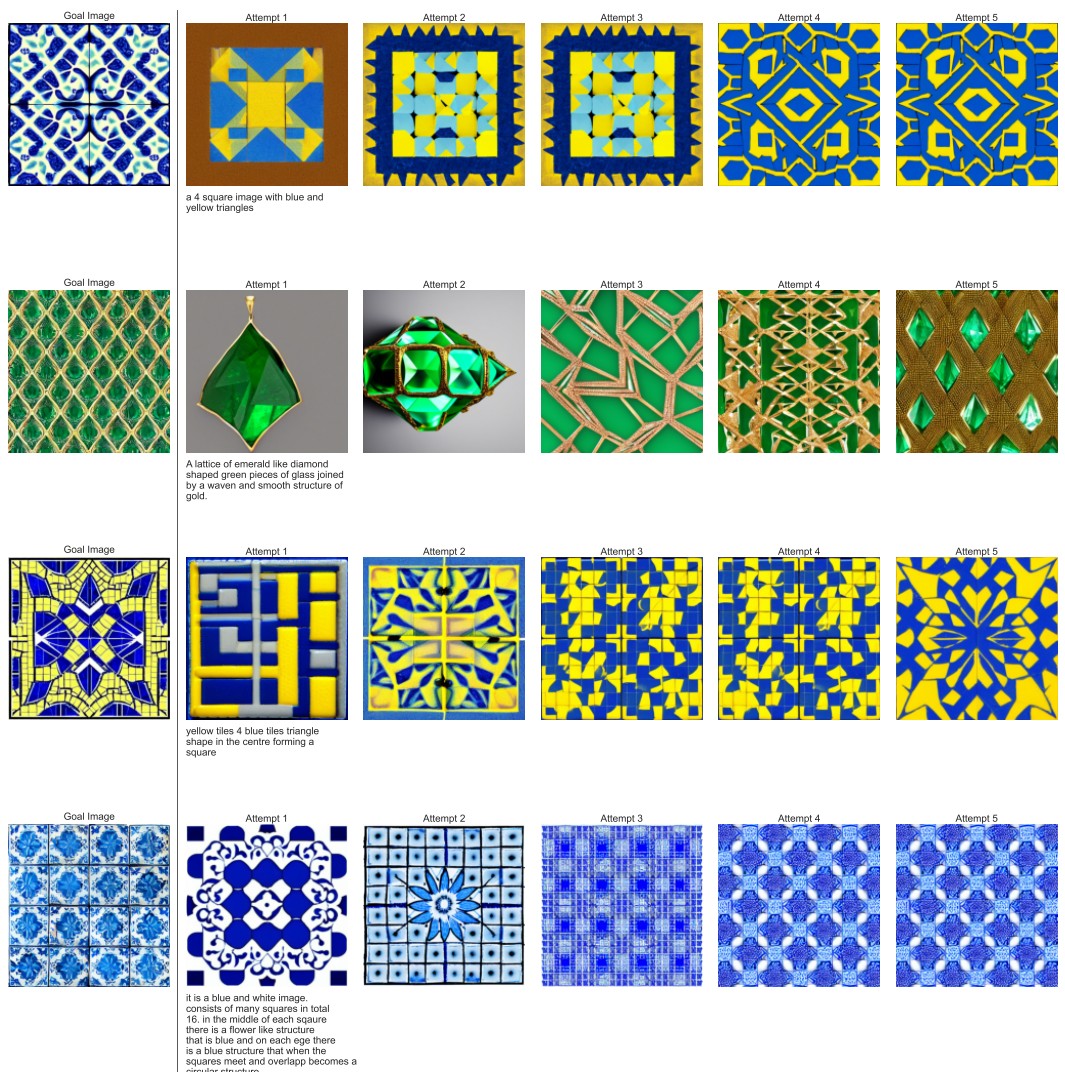

**Figure 13:** Users' attempts at using image steering to reproduce goal images of tile patterns that are difficult to articulate.

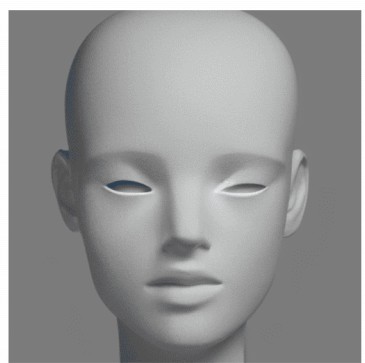

## Reference Image Preview (Round 1/2)

The image above is your reference image for this round (it may take a few seconds to load).

**What's next:** On the following screen, you'll see this same image again, along with a text box where you can enter your descriptions. You'll have multiple attempts to refine your description and get the AI-generated image to match this reference image as closely as possible.

**Figure 14:** Instructions for an image generation survey.

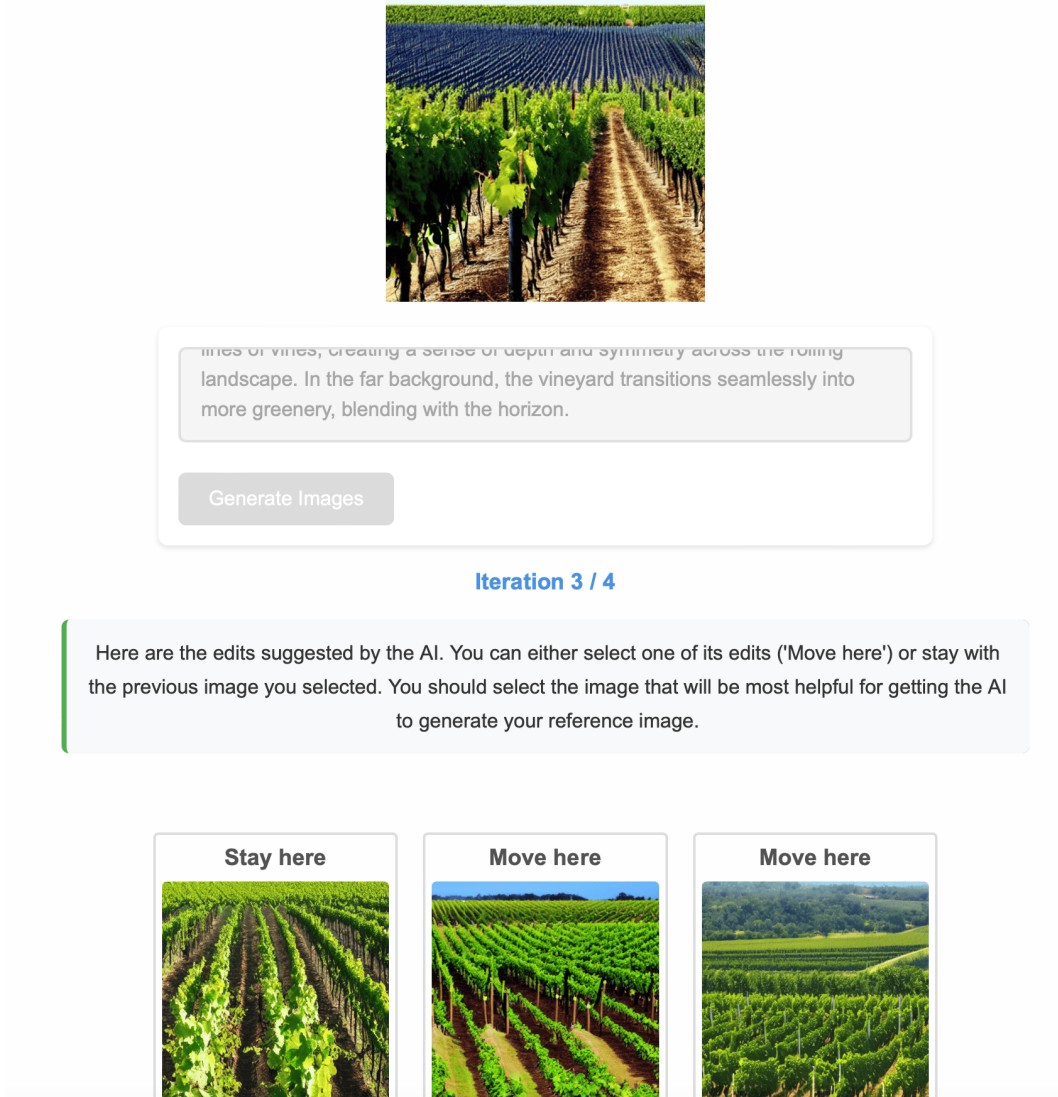

**Figure 15:** Example screen for an image generation survey with image steering.

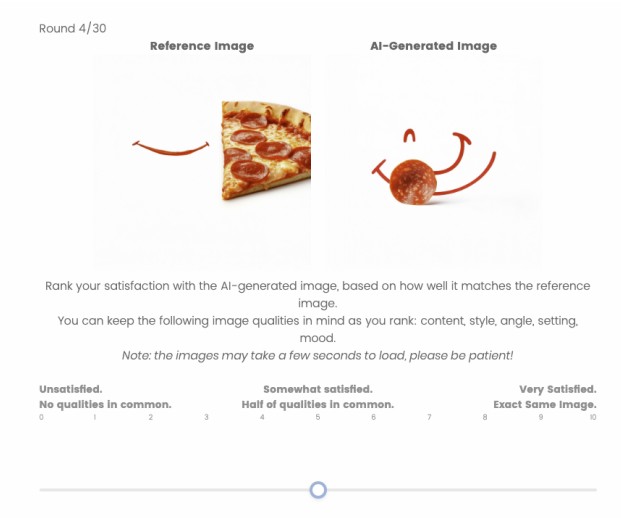

**Figure 16:** Example screen for the generated vs goal image similarity survey.

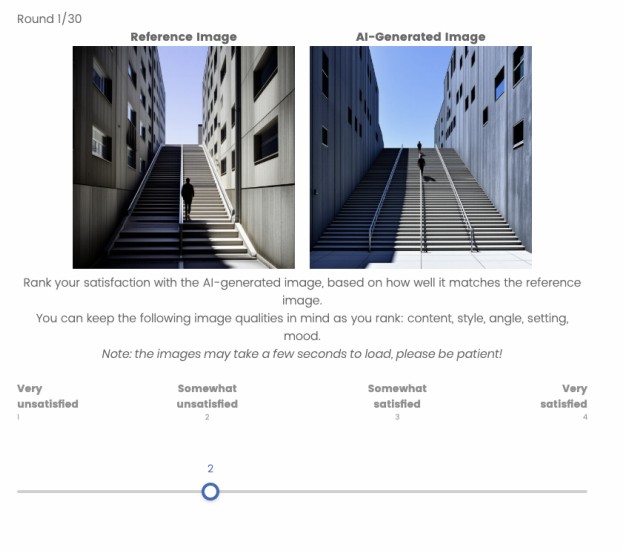

**Figure 17:** Example screen for the satisfaction rating survey.

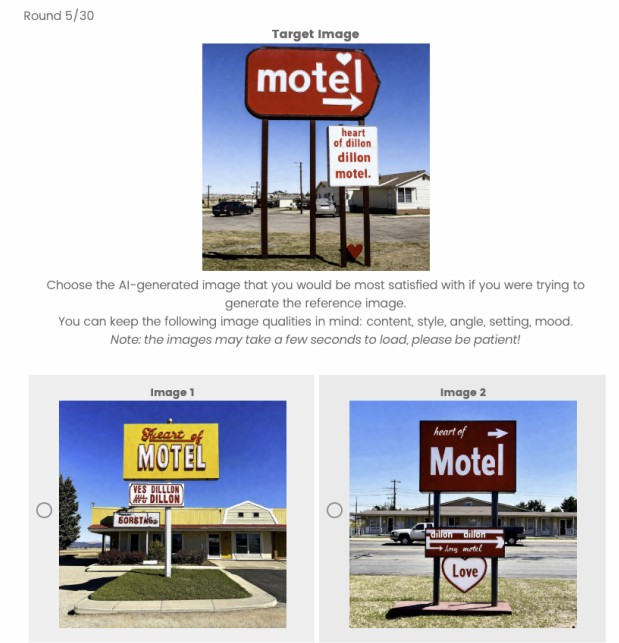

**Figure 18:** Example screen for the improvement rating survey.

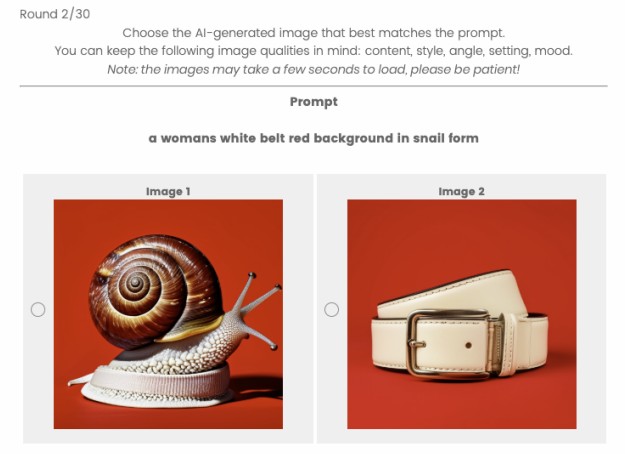

**Figure 19:** Example screen for the prompt-output misalignment survey.

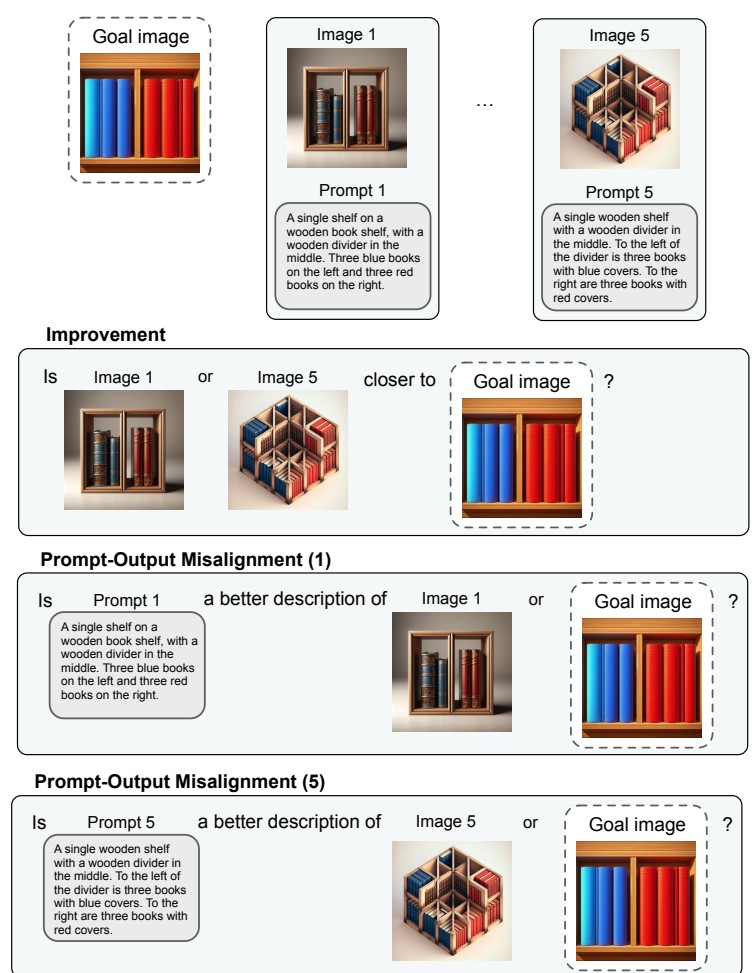

**Figure 20:** Metrics used for evaluating the steerability of text-to-image models. See Section 3 for more details.

