# OpenReview forum: "What's Producible May Not Be Reachable: Measuring the Steerability of Generative Models"
_NeurIPS.cc/2025/Conference — NeurIPS 2025 poster_

### Official Review · Reviewer_k56Q · 2025-06-13

**Clarity:** 2
**Significance:** 3
**Originality:** 3
**Rating:** 4
**Confidence:** 4

**Summary:**

The paper proposes a formal decomposition that separates a generative model's producible set from its steerability gap, gives a human-in-the-loop benchmark that samples a model's own output as a goal and asks users to reproduce it, and reports large-scale user studies on 10 text-to-image systems and 5 LLMs. Empirically, users fail to reach the targets: annotators judge 60% of image attempts unsatisfactory and prompt refinement helps only 62% of the time. For LLMs, "very satisfied" ratings drop to 17%. A CLIP-based predictor explains roughly half of the variance in steerability, and a simple "image-steering" interface that proposes visual variations yields more than a two-fold improvement over text prompting.

**Questions:**

1. I need clarity on the benchmark protocol. You cap each reproduction attempt at five prompts and forbid negative prompts or ControlNet style constraints. Why were those particular limits chosen, and how sensitive is the failure rate to relaxing them? A small ablation that doubles or halves the attempt budget, or that enables commonly used guidance knobs, would establish whether the headline 60 percent unsatisfied rate is intrinsic or policy driven.
2. The LLM study forbids any lexical overlap with the target text. That constraint is not typical of real usage, where reusing phrases is often encouraged to preserve semantics. Could you report satisfaction scores when overlaps are allowed, or justify why the overlap prohibition models an important practical scenario?

**Ethical Concerns:**

["NO or VERY MINOR ethics concerns only"]

**Final Justification:**

I've updated my score to a borderline accept. The rebuttal was clear and the authors acknowledged and correctly responded to all my questions.

**Limitations:**

discussed above

**Paper Formatting Concerns:**

Writing is generally clear. I didn't find any notable paper formatting issues.

**Quality:**

2

**Strengths And Weaknesses:**

Isolating steerability from producibility is conceptually useful; related work has evaluated controllability but typically conflates the two dimensions. The benchmark is sufficiently large to matter and the released data could become a reference point. That said, the decomposition is mathematically straightforward once the goal is stated, and the benchmark design (i.e., reproduce the model's own sample) is a narrow proxy for real user intent. Compared with recent evaluation surveys, the contribution is incremental on the measurement side and modest on the algorithmic side.
The framework is sound but largely descriptive; no theoretical guarantees or identifiability claims are offered. The predictor uses CLIP embeddings, yet CLIP correlates only 0.32 with human judgments, so relying on it to partition gains risks substantial error propagation. The experimental protocol omits potentially decisive variables: random seeds are hidden, negative prompts and control-net style constraints are disabled, and only five attempts are allowed. These design choices may artificially depress steerability and inflate the headline 60% failure rate. The LLM study forbids using words from the goal text, a restriction that is unlikely in real usage and may explain the low 17% satisfaction figure. No ablations quantify the sensitivity of results to these constraints.
The user studies are large, but the statistical treatment is thin. Aside from standard errors, there is no modelling of worker or item-level variance, no correction for repeated-measures correlations, and no preregistered stopping rule. The claim that image steering "more than doubles" improvement is true on DreamSim deltas, yet the baseline improvement is itself tiny, so the practical effect size is unclear. The blind-LLM baseline achieves 52% of human improvement with no goal awareness, implying that the benchmark rewards prompt lottery effects rather than systematic steering; this undercuts the core thesis.
The authors collected 18 k human ratings but provide no analysis of demographic effects or annotation consistency. Asking workers to replicate potentially copyrighted output raises legal issues. Improving steerability may also facilitate malicious uses; this is acknowledged only briefly.

---

> ### Author Rebuttal · Authors · 2025-07-28
>
> Thank you for your insightful review. Your comments were very constructive and have improved the paper. To summarize the main changes, we've added:
> - Analysis for different numbers of allowed rewrite attempts
> - Further comparison of CLIP/Dreamsim to human metrics
> - Clarification about experimental design
>
> > _You cap each reproduction attempt at five prompts and forbid negative prompts or ControlNet style constraints. Why were those particular limits chosen, and how sensitive is the failure rate to relaxing them? A small ablation that doubles or halves the attempt budget, or that enables commonly used guidance knobs, would establish whether the headline 60 percent unsatisfied rate is intrinsic or policy driven._
>
> It's a good idea to look at how sensitive results are to the number of attempts. We've added results below across different numbers of prompts, which we'll add to the paper:
>
> | Attempt | Percent unsatisfied | Dreamsim score (higher is better) |
> | - | - | - |
> | 1 | 0.64 (0.02) | 0.639 (0.008) |
> | 2 | 0.62 (0.02) | 0.661 (0.007) |
> | 3 | 0.56 (0.02) | 0.683 (0.007) |
> | 4 | 0.58 (0.02) | 0.684 (0.007) |
> | 5 | 0.60 (0.02) | 0.695 (0.007) |
>
> For all of the number of rounds, unsatisfactory rates are within 4% of the headline result (60% dissatisfaction). The paper considered five attempts because we found only marginal improvement after the third attempt. We also feel that five is a realistic number of times a user would try rewriting prompts. We completely agree that longer time horizons (e.g. multiple days of weeks) would be interesting, and we don't mean to suggest that people cannot improve over time. We'll emphasize these points in our revision.
>
> We focused on features that were shared across APIs, which is why we didn't include negative prompts or ControlNet style constraints. We did try an experiment (Table 6 in the appendix) where we allowed users to specify classifier-free guidance (CFG) settings in addition to text prompting. We could only test Stable Diffusion models for this because not all APIs had CFG. Users didn't show significant improvement with or without CFG (DreamSim similarity score, higher is better, SE in parentheses):
>
> | Attempt | Without CFG | With CFG |
> |-|-|-|
> | 1| 0.638 (0.020) | 0.664 (0.018) |
> | 2| 0.662 (0.020) | 0.677 (0.017) |
> | 3| 0.676 (0.015) | 0.667 (0.017) |
> | 4| 0.681 (0.015) | 0.703 (0.015) |
> | 5| 0.693 (0.015) | 0.682 (0.017) |
> | **Average** | 0.670 (0.008) | 0.679 (0.007) |
>
> We think testing more advanced features would be interesting. These mechanisms are all amenable to the framework in the paper, and we hope that future work will study additional features.
>
> > _The predictor uses CLIP embeddings, yet CLIP correlates only 0.32 with human judgments, so relying on it to partition gains risks substantial error propagation._
>
> Thanks for bringing this up. We wanted to clarify a potential confusion: the 0.32 CLIP correlation shows that the CLIP score between a user's prompt and the resulting image has a low correlation with the steerability score for that image. This suggests that CLIP scores can be high with poor steerability, e.g. if many images are consistent with a single prompt or if a user has many desiderata that are infeasible to convey in a single prompt. Meanwhile, the predictor in the decomposition in Section 4 uses CLIP not as a score but as an embedding to a larger model and uses it not for the user's reproduced image but for the goal image (the goal is to predict how steerable a given goal image is, so it's not a function of reproduced images). We find that these models achieve an R^2 of almost 0.50, explaining almost half of the variance in poor steerability. We realize the writing wasn't as clear as it should've been, and we'll update this in the main paper.
>
> When we use CLIP elsewhere in the paper, it's to compare its scores to human scores. We agree that it would be risky to rely on CLIP as a proxy for humans without benchmarking, so we use human annotations for our main experiments in Section 3. We've also included analysis below (which expands on results in the appendix) to compare human-based annotations to automated ones (based on CLIP and DreamSim scores), which finds that the automated annotations have high correlation with the human-based ones:
>
> | Model | Average human rating (10-point scale) | Average DreamSim | Average CLIP |
> | - | - | - | - |
> | DALL-E 2 | 2.97 | 0.52 | 0.75 |
> | DALL-E-3 | 4.08 | 0.62 | 0.78 |
> | Flux-dev | 4.66 | 0.70 | 0.83 |
> | Flux-1.1-pro-ultra | 4.34 | 0.66 | 0.82 |
> | Ideogram-v2-turbo | 4.27 | 0.66 | 0.83 |
> | Photon-flash | 4.84 | 0.68 | 0.85 |
> | SD3-large | 4.45 | 0.68 | 0.84 |
> | SD3.5-medium | 4.37 | 0.67 | 0.81 |
> | SD3.5-large-turbo | 4.13| 0.65 | 0.82 |
> | SD3.5-large | 4.48 | 0.67 | 0.82 |
> | **Correlation to average human rating** | --- | 0.969 | 0.904 |
>
> > _The claim that image steering "more than doubles" improvement is true on DreamSim deltas, yet the baseline improvement is itself tiny, so the practical effect size is unclear._
>
> We agree that the baseline improvements look small in DreamSim score, and that DreamSim units can be hard to interpret. For a more interpretable metric, note that the improvement rate for text steering is ~55% compared to ~72% for image steering. This is an improvement from 5% over chance to 22% over chance, which we feel is sizable.
>
> > _The LLM study forbids any lexical overlap with the target text. That constraint is not typical of real usage, where reusing phrases is often encouraged to preserve semantics. Could you report satisfaction scores when overlaps are allowed, or justify why the overlap prohibition models an important practical scenario?_
>
> Our goal with the LLM experiments is to capture one frequent use-case for LLMs in writing: to rewrite a passage for a user who can identify the style they want when they see it, but is unable to articulate what exactly that style is. For example, a student wishing to use an LLM to edit academic, technical writing may not get what they're looking for if they prompt an LLM to provide edits that match "academic, technical writing". However, even if they don't know exactly how to describe the writing they want, they are usually able to identify the examples they're happy or unhappy with when they see them. Our results found that this aspect of steerability is difficult for LLMs. We think this results adds to the conversation about LLM steerability: like you suggested, there are many uses where LLMs might be steerable (e.g. solving math problems), but in this focused aspect of writing to match a certain tone, they are less steerable. However, we don't aim to suggest that writing/editing is the only use-case for LLMs, and we will rewrite that section to emphasize these points.
>
> > _The blind-LLM baseline achieves 52% of human improvement with no goal awareness, implying that the benchmark rewards prompt lottery effects rather than systematic steering._
>
> This is a great point. The purpose of the blind LLM is to serve as a baseline. We want to clarify that the blind LLM does not do well, and it helps calibrate human improvement, which is already small. This could be seen as analogous to image classification experiments, where an image classifier that always predicts the majority class is included to show how well an uninformative model can perform. We feel the LLM is a useful baseline, but another option is to limit the number of human attempts to 1, so there is no lottery effect. This is an important point and we'll add it to our discussion.
>
> > _The authors collected 18 k human ratings but provide no analysis of demographic effects._
>
> While the data we recorded does not include demographics, we also recruited professional prompt engineers to perform the benchmark task via Upwork and compared them to the nonprofessional group. We found that professionals were better than the average steerer from the nonprofessional survey, but only slightly: final similarity scores were only 10% higher than for nonprofessionals. Moreover, we find that professional prompt engineers don't reliably improve between their first and last attempts, with even lower improvement rates than among the nonprofessional cohort.
>
> > _That said, the decomposition is mathematically straightforward once the goal is stated, and the benchmark design (i.e., reproduce the model's own sample) is a narrow proxy for real user intent._
>
> We completely agree that the decomposition is mathematically straightforward. We feel this simplicity is a strength: a simple decomposition is easier to interpret than a more complicated formula. You're also correct that the goal image is sampled from the same model used for steering. As discussed in Section 2, this ensures that steerability is independent of producibility: if a user can't reproduce the goal image, it's not about the quality of the model's images, just its steerability.
>
> > _Asking workers to replicate potentially copyrighted output raises legal issues. Improving steerability may also facilitate malicious uses; this is acknowledged only briefly._
>
> Thanks for raising this concern. We designed the benchmark with the goal of avoiding copyright issues: each image used in the benchmark is generated by a text-to-image model, and is not drawn from any external, copyrighted dataset. Still, as with any use of text-to-image models, it's possible that a model may produce a copyrighted image, and we will bring this up in our discussion. On the broader question of potential misuse, we agree that improved steerability could be dual use. That said, we believe that understanding and quantifying steerability is an important prerequisite for responsible deployment and mitigation planning. We'll expand on these points in our revision.

---

### Official Review · Reviewer_hQSC · 2025-06-29

**Clarity:** 3
**Significance:** 3
**Originality:** 4
**Rating:** 4
**Confidence:** 4

**Summary:**

This paper introduces a new framework for evaluating generative models by disentangling two key dimensions: producibility (what a model can generate) and steerability (how well a user can guide the model toward a desired outcome). The authors propose a benchmark task that samples an output from a model and asks users to reproduce it, enabling the measurement of steerability independently from producibility. Through large-scale user studies on text-to-image and language models, they find that current models, despite their high output quality, exhibit poor steerability. The paper further shows that steerability can be predicted using machine learning and can be significantly improved through alternative interaction mechanisms, such as image-based steering. These findings highlight the importance of steerability in real-world usability and call for a shift in how generative models are evaluated and developed.

**Questions:**

Human Prompting Variance in T2I Tasks:
Given that the T2I benchmark requires users to prompt a model based on a target image, how do the authors account for the high variance caused by differing levels of user familiarity with prompt engineering or specific model behaviors? Could the authors report or control for user background (e.g., prior experience with prompting or specific models)? A clearer characterization of user profiles and possibly stratified analysis would improve confidence in the benchmark’s robustness.

Lack of Iterative Editing in Steering Protocol:
Many modern T2I workflows involve multi-round editing or prompt refinement (e.g., iterative inpainting or variation selection). Why does the benchmark focus only on one-shot or limited prompt-based steering? Have the authors considered evaluating more realistic multi-turn interaction paradigms, and if not, could they discuss potential effects this simplification may have on the reported steerability performance?

Evaluation Metrics and Granularity of Assessment:
The use of CLIP similarity and DreamSim captures high-level alignment but may overlook fine-grained fidelity. Since steerability often depends on whether specific details (e.g., exact number of objects, spatial arrangements) can be reproduced, would the authors consider incorporating or comparing against MLLM-based evaluators (e.g., GPT-4o, Qwen2.5-VL) that could better judge localized attribute matching? Evidence that CLIP-based metrics align well with human or MLLM-based fine-grained evaluation would strengthen the current claims.

Necessity of Steerability Evaluation in LLMs:
One broader question is whether LLMs should be evaluated for steerability in the same way as T2I models. In my view, LLMs are already commercially successful largely because of their strong steerability in natural language interactions. Therefore, the practical need to rigorously benchmark LLM steerability may be less pressing. The authors could clarify what specific failure modes or gaps in LLM usability motivated their inclusion in this study, and what new insights this part of the benchmark contributes beyond prior understanding.

**Ethical Concerns:**

["NO or VERY MINOR ethics concerns only"]

**Final Justification:**

The rebuttal addressed my main questions by providing additional analyses on metric–human correlation, user background effects, and the rationale for the one-shot steering setup.

However, I still have some reservations about the GPT‑4o evaluation results, as the discrepancy with human ratings might be influenced by the prompt design for GPT‑4o.

While these concerns remain, I find the work’s exploration of steerability to be a meaningful and timely contribution. I therefore maintain my borderline accept recommendation.

**Limitations:**

yes

**Quality:**

3

**Strengths And Weaknesses:**

Strengths:

1. The paper presents a novel and valuable perspective on steerability in generative models, distinguishing it from the commonly emphasized producibility. The proposed framework and benchmark provide a concrete and actionable methodology for evaluating steerability, which is likely to inspire further research in this direction.

2. The authors conduct comprehensive experiments on both state-of-the-art text-to-image models and large language models, offering a broad and reproducible analysis.

Weaknesses:

1. The design of the T2I (text-to-image) steerability benchmark relies heavily on users' ability to craft effective prompts. Since users are shown a target image and asked to prompt the model to reproduce it, the outcomes are significantly influenced by users' familiarity with the model and prompt engineering experience, potentially introducing high variance into the evaluation.

2. Many T2I models benefit from multi-step or iterative editing to refine outputs. However, the current benchmark setup focuses on single-prompt-based steering and does not consider this interactive refinement process, which limits its realism.

3. The evaluation metrics, such as CLIP embedding cosine similarity and DreamSim, largely capture global semantic alignment rather than fine-grained fidelity. Yet, effective steerability often hinges on reproducing subtle details. Incorporating vision-language models like GPT-4o or Qwen2.5-VL as evaluators might provide more nuanced and reliable assessments.

---

> ### Author Rebuttal · Authors · 2025-07-28
>
> Thank you for your positive review of our paper. We're glad that you found the framework novel and valuable, and the experiments to be comprehensive and "likely to inspire further research in this direction."
>
> We respond to your comments below. **We hope our comments have addressed your questions. If not, please let us know if you have any more questions we can address in the follow-up.**
>
> > _The use of CLIP similarity and DreamSim captures high-level alignment but may overlook fine-grained fidelity. Since steerability often depends on whether specific details (e.g., exact number of objects, spatial arrangements) can be reproduced, would the authors consider incorporating or comparing against MLLM-based evaluators (e.g., GPT-4o, Qwen2.5-VL) that could better judge localized attribute matching? Evidence that CLIP-based metrics align well with human or MLLM-based fine-grained evaluation would strengthen the current claims._
>
> It's a good idea to compare automated alignment metrics with those of humans. We've included analysis below comparing human-based annotations to automated ones (based on CLIP and DreamSim scores), which finds that the automated annotations have high correlation with the human-based ones:
>
> | Model | Average human rating (10-point scale) | Average DreamSim | Average CLIP |
> | - | - | - | - |
> | DALL-E 2 | 2.97 | 0.52 | 0.75 |
> | DALL-E-3 | 4.08 | 0.62 | 0.78 |
> | Flux-dev | 4.66 | 0.70 | 0.83 |
> | Flux-1.1-pro-ultra | 4.34 | 0.66 | 0.82 |
> | Ideogram-v2-turbo | 4.27 | 0.66 | 0.83 |
> | Photon-flash | 4.84 | 0.68 | 0.85 |
> | SD3-large | 4.45 | 0.68 | 0.84 |
> | SD3.5-medium | 4.37 | 0.67 | 0.81 |
> | SD3.5-large-turbo | 4.13| 0.65 | 0.82 |
> | SD3.5-large | 4.48 | 0.67 | 0.82 |
> | **Correlation to average human rating** | --- | 0.969 | 0.904 |
>
> Correlations with human ratings are high for both DreamSim and CLIP. Based on your suggestion, we also considered using GPT-4o as an MLLM-based evaluator. Because of API costs, we collected similarity scores from GPT-4o using a sample of 500 responses. The average scores for the human ratings and GPT-4o ratings on this sample are below:
>
> | Model | Average GPT-4o rating | Average human rating |
> | - | - | - |
> | DALL-E 2 | 2.85 | 2.83 |
> | DALL-E 3 | 3.63 | 4.18 |
> | Flux-1.1-pro-ultra | 4.50 | 4.06 |
> | Flux-dev | 4.25 | 4.02 |
> | Ideogram-v2-turbo | 4.19 | 4.03 |
> | Photon-flash | 4.19 | 4.85 |
> | SD3-large | 4.19 | 4.54 |
> | SD3.5-large | 3.89 | 4.31 |
> | SD3.5-large-turbo | 3.55 | 4.24 |
> | SD3.5-medium | 4.74 | 4.39 |
> | - | - | - | - |
> | **Correlation to average human rating** | **0.67** | -- |
>
> So the MLLM-based evaluation also has a strong correlation with the human ratings, but not as strong as CLIP or DreamSim.
>
>
> > _Given that the T2I benchmark requires users to prompt a model based on a target image, how do the authors account for the high variance caused by differing levels of user familiarity with prompt engineering or specific model behaviors? Could the authors report or control for user background (e.g., prior experience with prompting or specific models)?_
>
> Great suggestion. While the crowdsourcing platform we use does not allow us to filter by prompting expertise, we also recruited professional prompt engineers to perform the benchmark task via Upwork and compared them to the nonprofessional group. We found that professionals were better than the average steerer from the nonprofessional survey, but only slightly: final similarity scores were only 10% higher than for nonprofessionals. Moreover, we find that professional prompt engineers don't reliably improve between their first and last attempts, with even lower improvement rates than among the nonprofessional cohort. We think the comparison between these two groups is quite interesting, and will emphasize it in the revised paper.
>
>
> > _Many modern T2I workflows involve multi-round editing or prompt refinement (e.g., iterative inpainting or variation selection). Why does the benchmark focus only on one-shot or limited prompt-based steering?_
>
> It's a good idea to consider multi-round editing or inpainting. However we faced a practical challenge: most of the models were only available via their APIs. Because our goal was to compare models, we focused on features that were shared across APIs, which is why we didn't include multi-turn interactions. There are many possible steering mechanisms and we don't mean to suggest that our paper considers all of them; only the common ones that are shared across APIs. We will clarify this in the paper.
>
> To understand whether users were struggling to steer because of lack of control to more advanced features, Table 6 in the Appendix includes an experiment where we allowed users to specify classifier-free guidance (CFG) settings in addition to text prompting. We could only test Stable Diffusion models for this because not all APIs had CFG. Users show no significant improvement with or without CFG (DreamSim similarity score, higher is better, SE in parentheses):
>
> | Attempt | Without CFG | With CFG |
> |-|-|-|
> | 1| 0.638 (0.020) | 0.664 (0.018) |
> | 2| 0.662 (0.020) | 0.677 (0.017) |
> | 3| 0.676 (0.015) | 0.667 (0.017) |
> | 4| 0.681 (0.015) | 0.703 (0.015) |
> | 5| 0.693 (0.015) | 0.682 (0.017) |
> | **Average** | 0.670 (0.008) | 0.679 (0.007) |
>
>
> > _One broader question is whether LLMs should be evaluated for steerability in the same way as T2I models... The authors could clarify what specific failure modes or gaps in LLM usability motivated their inclusion in this study, and what new insights this part of the benchmark contributes beyond prior understanding._
>
>
> Our goal with the LLM experiments is to capture one frequent use-case for LLMs in writing: to rewrite a passage for a user who can identify the style they want when they see it, but is unable to articulate what exactly that style is. For example, a student wishing to use an LLM to edit academic, technical writing may not get what they're looking for if they prompt an LLM to provide edits that match "academic, technical writing". However, even if they don't know exactly how to describe the writing they want, they are usually able to identify the examples they're happy or unhappy with. Our results found that this aspect of steerability is difficult for LLMs. We think this results adds to the conversation about LLM steerability: like you suggested, there are many uses where LLMs might be steerable (e.g. solving math problems), but in this focused aspect of writing to match a certain tone, they are less steerable. However, we don't aim to suggest that writing/editing is the only use-case for LLMs, and we will rewrite that section to emphasize these points.

---

> > ### Comment · Reviewer_hQSC · 2025-08-02
> >
> > Thank you for your detailed rebuttal and for providing the additional analyses. I still have some reservations about the GPT‑4o evaluation results — I suspect the discrepancy with human ratings might be related to the specific prompts used for GPT‑4o. Overall, I think exploring steerability is an important and meaningful effort, and I appreciate your work in this direction. I will maintain my original score.

---

### Official Review · Reviewer_WAEQ · 2025-07-02

**Clarity:** 3
**Significance:** 2
**Originality:** 3
**Rating:** 4
**Confidence:** 3

**Summary:**

Evaluating generative models is an open question. Many current metrics evaluate the quality and diversity of the generated outputs. However they miss a key characteristic that the authors call "steerability": the ability of a model to produce an output matching the user's goal. The authors propose to evaluate steerability separately from producibility by designing a simple benchmark task: can a human steer the model to generate a specific image (that the model is known capable of producing). Overall current models have poor steerability. Alternative ways of interacting with the model, such as using image samples instead of prompt rewrites, can improve steerability.

**Questions:**

1. Can you clarify how the prompt rewrites are made? Is this as part of a conversation or is it always from scratch? As I understand it, for image generation, each prompt attempt is made independently (the model is reinitialized between each attempt) but for text generation, the prompt rewrites is part of a conversation (Table 4). To me, doing a conversation might pose a problem: are you sure that the producibility of a given text is the same when using a single prompt (for the goal headline) and when doing a conversation (for the attempts)? Personally I'm not sure these texts have the same producibility. Please clarify this point in the paper.
2. Could you clarify the random image steering? Is it that you propose variations of the images with random perturbations in latent space, then the human chooses the best, you generate again variation of this chosen image, and repeat for several rounds?
3. Note: there is an hyperlink issue line 673.

**Ethical Concerns:**

["NO or VERY MINOR ethics concerns only"]

**Final Justification:**

The idea of studying the steerability aspect of generative models is interesting but I find the definition in the paper too challenging for current models. It limits many results of the paper to “steerability is hard”.

**Limitations:**

limitations discussed in the paper

**Paper Formatting Concerns:**

no concern

**Quality:**

3

**Strengths And Weaknesses:**

**Strengths**

1. The idea to decouple producibility and steerability as characteristics of generative models is interesting and well justified.
2. The benchmark task is simple and well executed.
3. Good quality experiments.
4. Proposition of methods to improve steerability.

**Weaknesses**

1. No discussion on the causes of poor steerability. Here is a key reference that is missing [1]. It shows that text-to-image models based on CLIP encoder (which is the case for at least Stable Diffusion, I have not checked for the other models used in the paper) share "systematic failures": they struggle with text containing e.g. negation and quantities.
To me, this explains the main results of the paper: 1. the process shown on Figures 1 and 10 would never lead to the expected result (except by chance), because by construction the generative models struggle with notions of e.g. negation and quantity, 2. the low similiarity scores and very low improvements from first attempt to last attempt (Figure 2), 3. the competitiveness of blind steering (Figure 3).
Studying the potential causes of poor steerability would have improved the benchmark, which in its current form basically gives the humans an impossible task. For instance, it would have been useful to inform the humans about the "systemic failures" of generative models beforehand, to improve the effect of the prompt rewrites.
2. Lack of a reference baseline. Because the goal is too hard, humans fail to accomplish the goal, no matter the model used (cf. Figure 2: similarities are quite low (mostly below 5 out of 10)). To me, they should not be compared to "Perfect similarity" but a different reference value which can be based on e.g. variations of the target image with the same prompt but different seed. It would give a better "ideal value" than a perfect 10/10.



[1] Tong et al. (2023), "Mass-Producing Failures of Multimodal Systems with Language Models."
(https://proceedings.neurips.cc/paper_files/paper/2023/hash/5d570ed1708bbe19cb60f7a7aff60575-Abstract-Conference.html)

---

> ### Author Rebuttal · Authors · 2025-07-28
>
> Thank you for your positive review of our paper. We're glad that you appreciated the ideas, methods, and experiments described in the paper. We respond to your comments below; to summarize the main changes, we've added:
> - New analysis that includes comparisons with different success metrics
> - Analysis of time taken as a possible cause of poor steering
> - Clarification about experimental design
>
> **We hope our comments have addressed your questions. If not, please let us know if you have any more questions we can address in the follow-up.**
>
> > _Here is a key reference that is missing [1]. It shows that text-to-image models based on CLIP encoder (which is the case for at least Stable Diffusion, I have not checked for the other models used in the paper) share "systematic failures"... Studying the potential causes of poor steerability would have improved the benchmark... For instance, it would have been useful to inform the humans about the "systemic failures" of generative models beforehand, to improve the effect of the prompt rewrites._
>
> Thank you for providing the reference -- we will add it to the paper. We think it brings up an important and relevant point: one factor of poor steerability is that CLIP's representations share systematic failures. It's tough to fully determine how much CLIP factors into the failures we see because most models have not released details about whether they use CLIP, but we will add a discussion of this point to our main results.
>
> We find that another important factor is that prompt-image alignment metrics (like CLIP) between a user's prompt and their reproduced photo is not very predictive of steerability: the correlation between CLIP and steerability scores is only 0.32. This suggests that intent is not well-captured by prompt-image alignment metrics, e.g. if many images are consistent with a single prompt or if a user has many desiderata that are infeasible to convey in a single prompt.
>
> We've also performed more analysis focusing on the role of time: we measured effort levels by measuring how long different kinds of steering (image vs text) take. We find that not only is image steering more effective, but it's also faster than text prompting (14 minute vs 10 minute median length). This suggests one cause of poor steerability: writing careful text prompts is time-consuming, and people might get tired:
>
> | Percentile | Text steering minutes | Image steering minutes |
> | - | - | - |
> | 20th percentile | 8.6 | 7.3 |
> | 40th percentile | 10.9 | 8.8 |
> | 60th percentile | 14.1 | 11.0 |
> | 80th percentile | 16.8 | 15.3 |
>
> Informing humans of the systemic failures of T2I models is an interesting idea. We recruited professional prompt engineers -- who all had experience with T2I models and their failures -- to perform the benchmark task via Upwork. We found that professionals were only slightly better than non-professionals: final similarity scores were only 10% higher than for nonprofessionals. Moreover, we find that professional prompt engineers don't reliably improve between their first and last attempts, with even lower improvement rates than among the nonprofessional cohort. This suggests that just being aware of the failures of T2I models doesn't result in significant improvement.
>
> We will add these analyses and more to the discussion in our paper.
>
> > _Because the goal is too hard, humans fail to accomplish the goal, no matter the model used (cf. Figure 2: similarities are quite low (mostly below 5 out of 10)). To me, they should not be compared to "Perfect similarity" but a different reference value._
>
> You make a great point: users struggle to steer when they use text as a mechanism. This motivated us to look at other mechanisms, and we found that although no mechanism results in perfect performance, improvement is possible -- for example, image steering (especially with RL) shows improvement.
>
> Additionally, the blind LLM in Figure 3 provides one kind of baseline: a steering mechanism that corresponds to randomly rewriting prompts without access to the goal image. The fact that this baseline achieves half of the human improvement helps calibrate the human performance against a baseline.
>
> Still, we don't expect users to exactly create their goal image, so we've also performed a new experiment that simulates users being equally happy with multiple goal images instead of one. Specifically, instead of measuring performance by distance to the goal image, we say that a steerer has succeeded if they produce any image within some distance. The table below shows the success rate for different thresholds:
>
> | Distance that counts as success (DreamSim) | Success rate |
> | - | - |
> | 0.50 | 0.87 |
> | 0.60 | 0.68 |
> | 0.70 | 0.41 |
> | 0.80 | 0.12 |
> | 0.90 | 0.00 |
>
> To get more than 50% success, the threshold needs to be 0.666 DreamSim score or lower. Empirically we find pairs of images at this threshold are not very similar. Steering remains challenging even with this relaxed definition of success.
>
>
>
> > _Can you clarify how the prompt rewrites are made? Is this as part of a conversation or is it always from scratch?_
>
> This is a good point, and our paper should have been more clear: Prompt rewrites are always from scratch. That is, a user sees their previous prompt, they're given the opportunity to edit it, and then submit it to the model. We completely agree that doing a conversation might pose a problem, which is why we have each prompt attempt made independently, for both text and image generation. We'll clarify this in the revision.
>
> > _Could you clarify the random image steering? Is it that you propose variations of the images with random perturbations in latent space, then the human chooses the best, you generate again variation of this chosen image, and repeat for several rounds?_
>
> Yes this is exactly right. Importantly, we choose the same number of rounds as in text steering so that the results are comparable. We provide more details in C.2 but we'll make this more clear and move it to the main text in the revision.
>
> > _Note: there is an hyperlink issue line 673._
>
> Good catch. This should read "Figure 3". We'll fix it in the revision.

---

> > ### Comment · Reviewer_WAEQ · 2025-08-04
> > **Response to rebuttal**
> >
> > Thank you for this detailed rebuttal.
> > However I still think that the benchmark task is too difficult (for current models at least) to really be meaningful. With this definition, steerability is too challenging. The relaxed definition of success proposed in the rebuttal might be a good starting point. On the other hand, I appreciate the main idea of studying a different aspect than producibility of generative models. I will maintain my rating.

---

### Official Review · Reviewer_QagF · 2025-07-02

**Clarity:** 4
**Significance:** 3
**Originality:** 3
**Rating:** 5
**Confidence:** 4

**Summary:**

The paper deals with evaluating generative model performance, seeking to define the measure of steerability - meaning, how well can a human guide the model to produce a specific result. This notion is separate from the more commonly depicted metrics, that do not separate the question of how much the produced output fits the description from the quality of the produced image or text. A framework of human assessment is suggested to tackle this task, and findings appear to show that all tested models have rather low steerability - meaning, only around half the time humans are satisfied with the results or find them similar to the intended goal. This measurement technique is applied both to image and text generation, and later used also to measure the effectiveness of steering image generation using image selection.

**Questions:**

While I understand the theoretic need for separating producibility from steerability, I wonder, if we repeated the same experiment with any type of goal image (e.g., a collection of stock photos), would we see the same metric values, or would we see lower satisfcation numbers because the images are not reproducible? I think this is an important comparison that would add to the paper's claims.

**Ethical Concerns:**

["NO or VERY MINOR ethics concerns only"]

**Final Justification:**

I voted to accept the paper initially, while raising some points that were adequately addressed by authors. I repeat my accept vote.

**Limitations:**

Limitations are adequately discussed.

**Paper Formatting Concerns:**

I commend you for adding standard errors in tables, please be more consistent in labeling them in the captions of all tables.

**Quality:**

3

**Strengths And Weaknesses:**

The paper has good insights into generative models, and captures a real gap between the focus of existing evaluation strategies and human usage of these models. The definitions of steerability are sound, and the experiment is valuable in terms of showing a true gap of following all user instructions - that generative models, especially text-to-image models, have a hard time with. The measurements are made using several models, and the metric is validated by showing that the superior method of using images to steer indeed achieves higher results in the metric.
However, the metric being heavily human-dependent, and thus expensive to compute or improve on. More work can be put into automating parts of the assessment and proving the automation does not hurt precision.
I also found the LLM assessment to be somewhat lacking. LLMs are rarely used in this scenario of reproducing a text in exactly the same way, but rather to generate text in a broader context. There's a very large difference between an image depicting the correct number of books, and a text using the exact same word. I think the metric does not fit the LLM scenario and this part should not have been included in the paper.

---

> ### Author Rebuttal · Authors · 2025-07-28
>
> Thank you for your positive review. We're glad you found the paper insightful and empirically valuable. We respond to your comments below; to summarize the main changes, we've added:
> - An experiment where users try to steer images toward stock images, and we find that the steering scores are indeed worse than in the original survey
> - More analysis of how similar the human ratings are to the automated ones
>
> > _However, the metric being heavily human-dependent, and thus expensive to compute or improve on. More work can be put into automating parts of the assessment and proving the automation does not hurt precision._
>
> This is a great suggestion. The main metrics in the paper rely on humans as annotators and steerers. While we use humans because measuring steerability involves measuring people, as you point out, this makes the metrics harder to automate. We've included analysis below, which expands on the analysis in the appendix to compare human-based annotations to automated ones (based on CLIP and DreamSim scores). We find that the automated annotations have high correlation with the human-based ones:
>
> | Model | Average human rating (10-point scale) | Average DreamSim | Average CLIP |
> | - | - | - | - |
> | DALL-E 2 | 2.97 | 0.52 | 0.75 |
> | DALL-E-3 | 4.08 | 0.62 | 0.78 |
> | Flux-dev | 4.66 | 0.70 | 0.83 |
> | Flux-1.1-pro-ultra | 4.34 | 0.66 | 0.82 |
> | Ideogram-v2-turbo | 4.27 | 0.66 | 0.83 |
> | Photon-flash | 4.84 | 0.68 | 0.85 |
> | SD3-large | 4.45 | 0.68 | 0.84 |
> | SD3.5-medium | 4.37 | 0.67 | 0.81 |
> | SD3.5-large-turbo | 4.13| 0.65 | 0.82 |
> | SD3.5-large | 4.48 | 0.67 | 0.82 |
> | **Correlation to average human rating** | --- | 0.969 | 0.904 |
>
> We also tried comparing human ratings to MLLM-based ratings like GPT-4o, which also found high correlation (0.67), although not as high as for DreamSim or CLIP (see response to hQSC).
>
> Replacing human steerers with automated methods is trickier, because there is less evidence that large multimodal models can capture human behavior with high fidelity. Still, in our image steering experiments, we use a simulated human as part of an RL method to learn the images to suggest to people. While our results show that the simulated human is accurate enough to result in improvement over a baseline, we think more work is needed in order to assess whether automated steering can be a good proxy metric for human steering.
> > _While I understand the theoretic need for separating producibility from steerability, I wonder, if we repeated the same experiment with any type of goal image (e.g., a collection of stock photos), would we see the same metric values, or would we see lower satisfcation numbers because the images are not reproducible? I think this is an important comparison that would add to the paper's claims._
>
> This is a great suggestion. Based on your suggestion, we ran an experiment during the rebuttal period where the goal images were randomly sampled stock images (so they weren't necessarily producible by the model), and users were prompted to steer DALL-E 3 toward them. Across 130 attempts, we found that the similarity scores (based on CLIP) were indeed uniformly smaller than the model's scores in the main survey (when the goal image was sampled from the model's producible set):
>
> |  | Overall average similarity | First attempt similarity | Last attempt similarity |
> | - | - | - | - |
> | Goal image is stock image | 0.727 | 0.731 | 0.726 |
> | Goal image is from model's producible set | 0.784 | 0.778 | 0.778 |
>
> For reference, every model's CLIP score in the main survey was in the range of 0.75 - 0.85, so the stock image setting would be worse than any of the 10 models in the survey. We agree this comparison is important, and we believe it will strengthen the paper.
>
> > _I also found the LLM assessment to be somewhat lacking. LLMs are rarely used in this scenario of reproducing a text in exactly the same way, but rather to generate text in a broader context... I think the metric does not fit the LLM scenario and this part should not have been included in the paper._
>
> We completely agree that our metrics are a more natural fit for the image models than LLMs, which is why we focus on image models. Our goal with the LLM experiment is to capture how LLMs are often used in writing: to rewrite a passage so it looks a certain way but without being able to verbalize what exactly the final text should be. We agree with your suggestion: we'll move the LLM experiments to the appendix, and spend more of the main text focusing on analysis.
>
> > _I commend you for adding standard errors in tables, please be more consistent in labeling them in the captions of all tables._
>
> Thank you. We will add more labels in the captions of the tables to point to the standard errors.

---

> > ### Comment · Reviewer_QagF · 2025-08-04
> > **Thank you for the clarifications**
> >
> > Thank you for the clarifications and additions to the paper, I hope to see the paper accepted.

---

### Comment · Area_Chair_8HsW · 2025-08-05
**Reminder: end of author-reviewer discussion period**

Dear reviewers,

This is a reminder that the end of author-reviewer discussion period is near. Please do carefully read all other reviews and the author responses; and discuss openly with the authors, especially on your own questions that the authors addressed.

Thank you.

---

### Note · Authors · 2025-08-12

We thank the reviewers for engaging so closely with our paper throughout the review and rebuttal periods. To summarize, we were inspired by the reviewer suggestions to add the following experiments and analyses during the rebuttal period (described in more detail in responses to the reviewers):
- A new exercise where users are instructed to steer models toward stock images. As predicted by theory, the steering scores in the original experiment (where images come from model producible sets) are better than in the experiment with stock images
- New analysis that uses a more relaxed success metric to measure steerability, which reaffirms the poor steerability metrics
- Analysis of time taken as a possible cause of poor steering
- More analysis of how similar the human ratings are to automated ratings (CLIP/Dreamsim) and multimodal models (GPT-4o)
- Analysis for different number of allowed rewrite attempts, which finds similar satisfaction rates across attempts
- Further writing clarifications (role of blind LLM as a baseline, difference between CLIP predictive experiments, preventing copyright material, etc.)

We thank all the reviewers for the feedback and suggestions. These additions have been instrumental for further strengthening the paper!

---

### Decision · Program_Chairs · 2025-09-17

**Decision:**

Accept (poster)

**Comment:**

The paper argues that evaluating generative models should separate producibility from steerability, with the latter defined as how well can human guide the model to produce a specific result, as opposed to what the model can generate. It proposes a simple, general benchmark: sample an output from a model’s own producible set and ask users to steer the same model to reproduce it. The authors introduced a framework of human assessment for both image and text generation. Through large-scale user studies, they find that only around half the time humans are satisfied with the results or find them similar to the intended goal. Further, the paper shows that steerability can be predicted using CLIP-based predictor and can be improved through the use of image samples instead of prompt rewrites.

The work is original in its steerability focus and framework, and practically relevant. The benchmark is simple yet revealing, the human studies are large, and the proposed image-steering mechanism shows meaningful gains. The author addressed many important points in the rebuttals, especially on the meaningfulness of the benchmark, clarity of protocol, evaluation, variance of user familiarity with prompting,. While there are still some room for improvements, I believe that the papers’ contribution on the timely focus, formulation, framework, evaluation protocols, and its actionable insights would be valuable to the community.

__Strengths__
- Isolating steerability from producibility is sound, useful, interesting, and timely
- The proposed framework and benchmark are simple, actionable, sufficiently large, and could become a reference point
- Experiments are comprehensive and showing the true gap of following all user instructions
- The authors further showcase additional method of using images to steer to achieve higher results in steerability

__Weakness__
- The framework's definition of steerability is too challenging to be meaningful. In the rebuttal, the authors propose a relaxed, automated measure, and the resulting image pairs still appear insufficiently similar. While I agree that this dissimilarity might signify that steering remains challenging, it could be that the dissimilarity is due to the fact that the DreamSim measurement is not emphasizing fine-grained fidelity, thus limiting the relaxation irrelevant to the steerability issue. Exploring a more suitable relaxation—e.g., via MLLM-based evaluation—could enhance the impact of this work.
- The evaluation relies heavily on human prompting, which introduces variance. In addition, the utilization of the detailed PixelProse captions in the framework, also impose strong prompting requirements. This suggests that the current benchmark still leaves room for future improvements, e.g. on limiting the human prompting variance by rendering the prompting requirement constrained.